# Apelin modulates inflammation and leukocyte recruitment in experimental autoimmune encephalomyelitis

Hongryeol Park [1] ✉, Jian Song [2], Hyun-Woo Jeong [1], Max L. B. Grönloh [3], Bong Ihn Koh [1], Esther Bovay[1], Kee-Pyo Kim [4], Luisa Klotz [5], Patricia A. Thistlethwaite [6], Jaap D. van Buul [3], Lydia Sorokin [2] & Ralf H. Adams [1] ✉

Demyelination due to autoreactive T cells and inflammation in the central nervous system are principal features of multiple sclerosis (MS), a chronic and highly disabling human disease affecting brain and spinal cord. Here, we show that treatment with apelin, a secreted peptide ligand for the G protein-coupled receptor APJ/Aplnr, is protective in experimental autoimmune encephalomyelitis (EAE), an animal model of MS. Apelin reduces immune cell entry into the brain, delays the onset and reduces the severity of EAE. Apelin affects the trafficking of leukocytes through the lung by modulating the expression of cell adhesion molecules that mediate leukocyte recruitment. In addition, apelin induces the internalization and desensitization of its receptor in endothelial cells (ECs). Accordingly, protection against EAE major outcomes of apelin treatment are phenocopied by loss of APJ/Aplnr function, achieved by EC-specific gene inactivation in mice or knockdown experiments in cultured primary endothelial cells. Our findings highlight the importance of the lung-brain axis in neuroinflammation and indicate that apelin targets the transendothelial migration of immune cells into the lung during acute inflammation.

Multiple sclerosis (MS) is a complex and chronic autoimmune disease with unknown etiology. The prevailing view is that the pathogenesis of MS involves autoreactive T cells, but also B lymphocytes, leading to immune responses against myelin or other antigens, inflammation, demyelination, and neuronal damage[1,2]. Experimental autoimmune encephalomyelitis (EAE), evoked by immunization with myelin proteins, is a widely used animal model for MS, which has been used to characterize the migration of autoreactive T cells across the blood–brain barrier and the demyelination of nerve axons[3,4]. EAE provides valuable insight into the early stages of immune cell activation and leukocyte extravasation across cerebral blood vessels that is relevant to MS and allows the explorative analysis of treatment options for acute MS and disease-modifying therapies. It is also established that the generation of encephalitogenic T cells capable of penetrating the blood-brain barrier involves activation steps in lymph nodes, spleen, intestine, and lung[5–9].

Endothelial cells (ECs), which form the innermost lining of the vascular network, are central to inflammatory processes inside and outside of the central nervous system. In the brain, the initial loose interaction of circulating leukocytes with the activated endothelium is

[1]Max Planck Institute for Molecular Biomedicine, Department of Tissue Morphogenesis, Münster, Germany. [2]Institute of Physiological Chemistry and Pathobiochemistry and Cells-in-Motion Interfaculty Centre (CIMIC), University of Münster, Münster, Germany. [3]Vascular Cell Biology Lab, Department of Medical Biochemistry, Amsterdam UMC, and Section Molecular Cytology at Swammerdam Institute for Life Sciences, Leeuwenhoek Centre for Advanced Microscopy, University of Amsterdam, Amsterdam, The Netherlands. [4]Department of Medical Life Sciences, College of Medicine, The Catholic University of Korea, Seoul, Republic of Korea. [5]Department of Neurology, University of Münster, Münster, Germany. [6]Division of Cardiothoracic Surgery, University of California, San Diego, CA, USA. ✉e-mail: Hongryeol.Park@mpi-muenster.mpg.de; ralf.adams@mpi-muenster.mpg.de

mediated mainly by integrin binding to vascular cell adhesion molecule 1 (VCAM-1) on the endothelial surface[10,11]. Chemokine signals presented on the luminal endothelial surface promote the activation of leukocyte integrins and, thereby, high affinity binding to endothelial adhesion molecules such as ICAM-1 (Intercellular Adhesion Molecule 1 or CD54), which induces arrest of the immune cells prior to transendothelial migration (TEM). To transmigrate through the endothelium and enter into the adjacent tissue, leukocytes migrate mainly paracellularly through endothelial junctions but, in some cases, also transcellularly through the cell body[12].

Recent work indicates that inflammatory processes can be modulated by apelin, which is a peptide ligand for the G protein-coupled receptor (GPCR) APJ/Aplnr. Apelin is proteolytically processed, converting a larger 77-amino acid (aa) (in humans) preproprotein into various shorter products including a biologically active 13-aa isoform[13]. This short peptide is further activated by a pyroglutamyl modification at its N-terminus, which generates the highly potent [Pyr1]-apelin-13 (termed A13 in the article). Apelin, the related peptide ligand apela/elabela, and their common receptor APJ/Aplnr regulate cardiovascular morphogenesis during development, but there is increasing evidence linking these molecules to cancer and other pathobiological processes including inflammation-related diseases[14–16].

In the present study, we show that administration of A13 delays and suppresses the development of EAE in mice. Analysis of apelin receptor expression by single cell RNA sequencing (scRNA-seq) and genetic labeling indicate that the GPCR is largely absent from adult brain endothelium, whereas expression is prominent in the lung and a few other peripheral organs. We find that A13 interferes with leukocyte transendothelial migration and the formation of immune cell clusters in the lung, a step that was previously shown to endow T cells with the capacity to enter the central nervous system[8,17–19]. Our data also indicate that apelin treatment induces desensitization and internalization of the receptor APJ/Aplnr and, accordingly, major outcomes of A13 administration are phenocopied by EC-specific inactivation of the *Aplnr* gene in adult mice or siRNA-mediated gene knockdown in cultured cells.

## Results

### A13 suppresses development of experimental autoimmune encephalomyelitis

Mice were immunized with myelin oligodendrocyte glycoprotein peptide (MOG35-55 and injected intraperitoneally (i.p) with either A13 or vehicle (PBS) every other day commencing on the day of immunization (Fig. 1a). Disease onset and severity were assessed daily and brains were removed for immunofluorescence and/or flow cytometry analyses on day (D) 16, corresponding to peak disease severity in PBS-treated mice. EAE incidence was significantly lower in A13-treated animals (Fig. 1a). While EAE symptoms were detected in most of vehicle-treated animals by day (D) 13, less than 20% of mice in the A13 group developed symptoms by D16. Furthermore, the average disease score was significantly lower in A13-treated mice (Fig. 1a), which continued to show lower disease severity until D30. Consistent with these findings, immunofluorescence staining of brains of A13-treated mice at D16 showed a lack of CD45+ immune cell infiltration (Fig. 1b, c). Expression of the cell adhesion molecule ICAM-1 in the endothelium, an indicator of the inflammatory response[20], was also strongly reduced in A13 brain samples (Fig. 1b). Flow cytometry confirmed substantially lower numbers of CD4+ T cells, CD8+ T cells, Th1 helper cells, and IL-17-secreting (Th17) helper cells, important disease drivers in MS and EAE[21,22], in EAE brains and spinal cords at D16 after A13 administration (Fig. 1e and Supplementary Fig. 1a, b).

Previous work has shown that the development of EAE pathology involves the reprogramming of T cells from an activated into a migratory mode in the lung, which is required to provide the cells with the capacity to enter the CNS[8,17,18]. Strikingly, immunofluorescence staining of tissue sections revealed changes in immune cell

recruitment to the lung in A13-treated mice. While the initial accumulation of CD45+ immune cells and cluster formation was substantially delayed by A13 at D7, the resolution of these clusters, visible in vehicle-treated controls at peak EAE, occurred also significantly slower, leading to a much larger number of CD45+ foci in A13 lungs at D16 (Fig. 1d; Supplementary Fig. 2a, b). Likewise, flow cytometry indicated higher numbers of CD4+ T cells, CD8+ T cells, Th1 and Th17 in lungs of A13-treated mice at peak EAE (D16) but not at disease onset (D11) (Fig. 1f). The weight of freshly isolated lungs was significantly higher in A13 treated mice than vehicle controls at peak EAE, but a smaller difference was already seen at D11 (Supplementary Fig. 3a). As these data indicated that the trafficking of immune cells through the lung is compromised by A13, we used flow cytometry to measure total CD45+ immune cells in peripheral blood. The number of CD45+ and different T cell populations in peripheral blood were highest at disease onset (D11) in the control EAE group, and were substantially lower in A13-treated EAE animals at the same time point (Supplementary Fig. 3b). Moreover, FACS analysis of the mediastinal lymph node, which is draining lymph and immune cells from the lung and other nearby organs, shows that A13 treatment leads to significant reduction of multiple immune cell populations at D11 but no longer at peak EAE (D16) (Supplementary Fig. 1a, b). In contrast, immune cells in the spleen showed no significant differences between A13 and vehicle control. Moreover, analysis of inguinal lymph nodes (near the site of MOG35-55 injection) at D7 also showed no differences in immune cell content, indicating that the initial immune response is not compromised by A13 (Supplementary Fig. 1c). Taken together, these data indicate that A13 treatment affects immune cell trafficking and thereby reduces the entry pathogenic cells into the central nervous system, which results in delayed EAE onset and reduced disease severity.

Next, we addressed the question whether A13 acts in an organ-specific fashion. While apelin receptor is broadly expressed by endothelial cells (ECs) in the developing mouse[23], expression in the adult is more restricted and, as previously published scRNA-seq data show[24], confined to certain organs such as heart, gut, kidney, testis and lung (Supplementary Fig. 4a). To validate this data, we employed *Aplnr-CreERT2* transgenic mice in a *R26-mTmG* Cre reporter background, which results in expression of green fluorescent protein (GFP) by *Aplnr*+ cells upon tamoxifen injection. Shortly after tamoxifen administration to adult mice, GFP was detectable throughout the microvasculature of the adult heart and kidney, but was limited to only a few vessels in spleen and liver (Supplementary Fig. 4b). The same strategy, analysis of *Aplnr-CreERT2*-expressing cells shortly after tamoxifen administration, labeled lung microvascular ECs in early postnatal stages but also in adult (8-week-old) and aged (78-week-old) mice (Supplementary Fig. 4c), respectively. In contrast, *Aplnr* expression was largely absent in adult brain, whereas GFP signal was readily detectable in postnatal brain endothelium where it also decorated a subset of Olig2+ oligodendrocyte-lineage cells (Supplementary Fig. 5a–d). Downregulation of *Aplnr* transcripts in adult relative to postnatal brain ECs is also supported by scRNA-seq data[25] and transcripts for the GPCR were also not upregulated in EAE (Supplementary Fig. 5e). These results argue that A13 is unlikely to directly affect brain endothelium in EAE, but, instead, might act through ECs in the lung or other organs.

We also analyzed intestine because this organ shows endothelial *Aplnr* expression in the adult (Supplementary Fig. 4a) and is an important site of immune surveillance with established relevance for MS and EAE[9,26–28]. Analysis of intestinal sections, however, revealed no significant changes in tissue morphology or the abundance of CD45+ immune cells in A13-treated EAE mice (Supplementary Fig. 6a, b).

### A13 reduces the endothelial inflammatory response

Next, we investigated how A13 affects the accumulation of immune cells in the lung. Previous studies have shown that A13 treatment

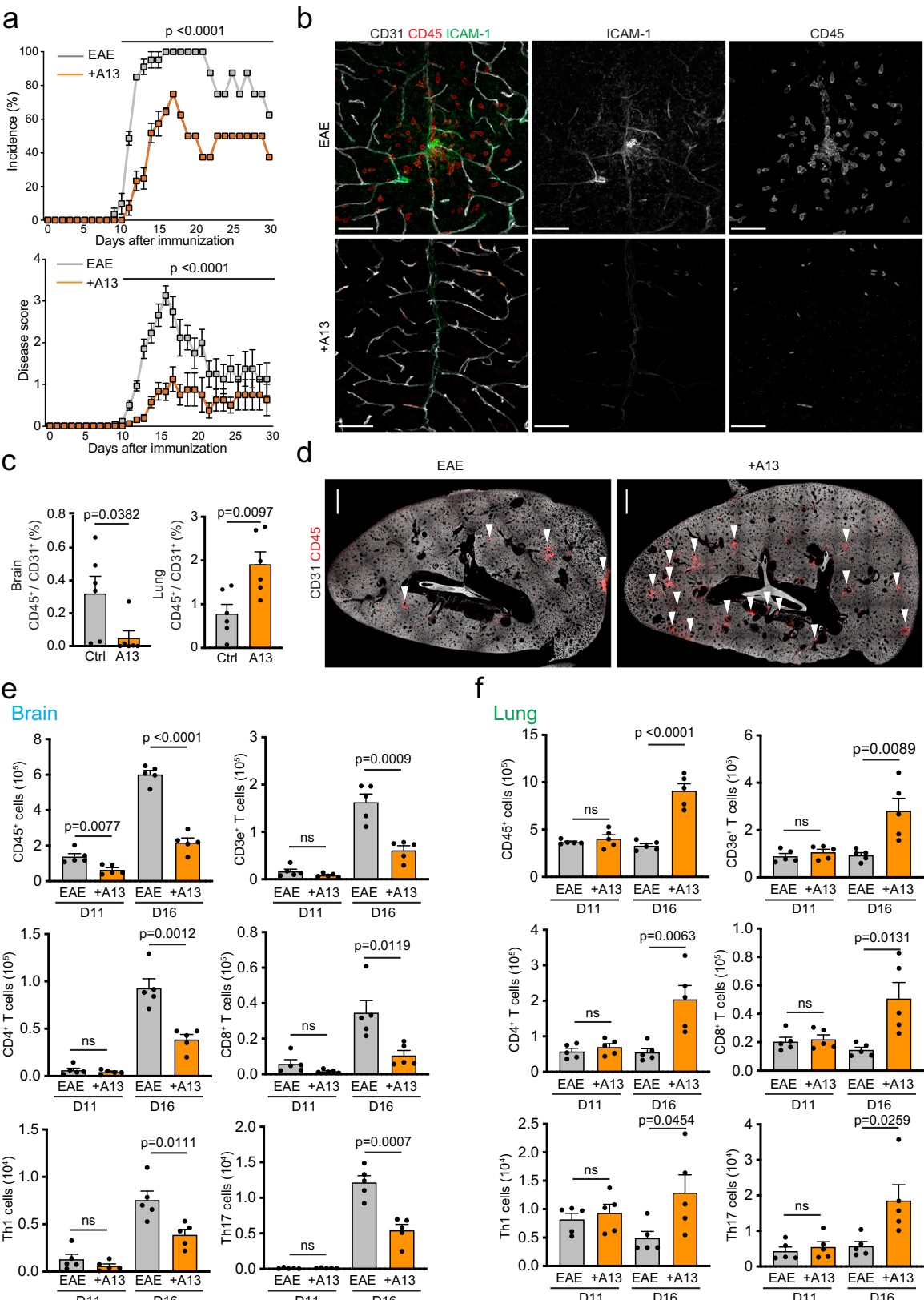

attenuates lipopolysaccharide (LPS)-induced acute lung injury[29–31]. To limit systemic effects and induce localized inflammation in the lung, we administered LPS (100 μg/40 μl) intranasally either alone or in conjunction with A13 (100 nM). Immunofluorescence staining revealed that A13 reduces the number of CD45+ cells in the lungs of treated mice relative to non-treated mice at 18 hours after LPS challenge

(Supplementary Fig. 7a). Four hours after treatment, flow cytometry showed that A13 strongly reduces the number of immune cells in bronchoalveolar lavage fluid (BALF) relative to controls, while immune cell numbers in peripheral blood were not significantly changed (Supplementary Fig. 7b and c). Consistent with the short A13 half-life[32], differences between treated and non-treated mice were less apparent

**Fig. 1 | A13 ameliorates EAE development and increases immune cell accumulation in the lung. a** Time course of MOG$_{35-55}$ immunization and A13 administration and analysis. Female C57Bl/6 mice aged 8-12 weeks received intraperitoneal (i.p.) injections of A13 or vehicle every other day commencing on the day of immunization (D0). Incidence rate and disease scores are shown up to day 30 (D30). Error bars, s.e.m. Two-way ANOVA. *n* = 27 mice/group (D0-D12, 4 independent experiments), 21 mice/group (D13-D15, 3 experiments), 14 mice/group (D16, 2 experiments), 8 mice/group (D17-D30, 1 experiment). **b** Representative confocal images showing sagittal sections of brain cortex from vehicle or A13-treated (+A13) mice at peak EAE (D16) immunostained for CD31 (white), ICAM-1 (green) and CD45 (red). Scale bar, 50 μm. Disease score, vehicle=3, +A13 = 1. **c** Quantification of immune cells infiltrating brain cortex or lung per vascular area at D16 (*n* = 6 mice each). Error bars, s.e.m. Two-sided Student's *t*-test. **d** Overview confocal images showing lung lobe sections at D16 EAE from vehicle control and A13-treated mice immunostained for CD31 (white) and CD45+ (red) immune cell clusters (arrowheads). Scale bar, 1 mm. Quantitation of CD45⁺ cells, CD3e⁺ T cells, CD4⁺ T cells, CD8⁺ T cells, Th1 cells, and Th17 cells in brain (**e**) and lung (**f**) by flow cytometry at EAE onset (D11) and peak (D16). Error bars, s.e.m. Two-sided Student's *t*-test. 5 mice per group.

at 24 hours after treatment (Supplementary Fig. 7b). Thus, a single treatment with A13 is sufficient to reduce acute immune cell recruitment to the lung.

Given that *Aplnr* expression is restricted to ECs, we investigated whether A13 might affect transendothelial migration of leukocytes. The cell surface glycoprotein ICAM-1 is expressed by ECs but also by epithelial and some immune cells. In ECs, ICAM-1 enables integrin-mediated stable adhesion of leukocytes and promotes inflammatory processes[33]. It is also known that endothelial ICAM-1 expression is strongly increased by inflammation and flow-induced shear stress[33,34]. Confocal imaging of immunostained tissue sections revealed that endothelial but not epithelial ICAM-1 staining intensity was significantly lower in A13-treated lungs relative to vehicle control at peak EAE (Fig. 2a). qPCR analysis of extracted RNA from whole lungs and from FACS-isolated ECs confirmed the downregulation of *Icam1* transcripts after A13 treatment (Fig. 2b). In addition, transcripts encoding chemokine ligand 2 (*Ccl2*) and interleukin 6 (*Il6*), important mediators of inflammation, were also downregulated in A13-treated lungs (Fig. 2b). In contrast, transcripts for the cell adhesion molecules VE-cadherin (*Cdh5*) and Claudin 5 (*Cldn5*), and for Gap Junction Protein Alpha 4 (*Gja4*), which mediate EC-EC junctional interactions, were not significantly altered. Interestingly, the endogenous expression of apelin (*Apln*) was also significantly reduced by A13, whereas *Aplnr* transcript levels were slightly decreased in whole lung without a significant change in sorted ECs (Fig. 2b). These results indicate that A13 treatment lowers endothelial ICAM-1 in the pulmonary endothelium but also reduces the expression of apelin itself and of cytokines that promote tissue inflammation.

As fundamental endothelial physiological functions, such as vascular permeability, are influenced by fluid shear stress, we investigated whether A13 treatment alters the response to shear (15 dyn/cm² for 18 hours) in cultured human umbilical cord venous endothelial cells (HUVECs) by bulk RNA sequencing (Fig. 2c-e). As expected, shear stress-induced a strong upregulation of the transcription factors Krüppel-like Factor 2 (*KLF2*) and *KLF4*, which was not affected by A13 (Supplementary Fig. 8). Apelin (*APLN*) was downregulated by fluid shear stress, whereas levels of apelin receptor transcripts (*APLNR*) were strongly increased, consistent with previous findings[35]. *APLNR* expression was further enhanced by A13 treatment (Supplementary Fig. 8). Importantly, shear stress-induced increases in the transcripts for ICAM-1 and VCAM were suppressed by A13 (Fig. 2c, e; Supplementary Fig. 8). A13 also reduced the expression of E-selectin (*SELE*), a molecule that enables the initial rolling and loose attachment of leukocytes from the bloodstream in peripheral tissues, but had no effect on the related P-selectin molecule (*SELP*) (Fig. 2c, e; Supplementary Fig. 8). Other regulators of the inflammatory response, including interleukin 6 (*IL6*), interleukin 8 (*IL8*), interleukin 1α (*IL1A*) and CXCL2, that have been reported to be upregulated by shear stress in human umbilical vein endothelial cells (HUVECs)[36,37], were downregulated by A13 to levels seen without flow (Fig. 2c, e; Supplementary Fig. 8). Gene set enrichment analysis (GSEA) confirmed that gene sets associated with the inflammatory response, tumor necrosis factor α (TNFα) signaling, IL6-STAT3 and IL2-STAT5 signaling were downregulated in response to A13 (Fig. 2d). These results argue that A13 affects leukocyte extravasation through gene expression changes in ECs.

## A13 suppresses inflammatory gene expression in Aplnr+ ECs, but not T cells

To systematically investigate EAE and A13-induced changes in different cell populations of the lung in vivo, we performed scRNA-seq analysis of untreated controls (8 to 9-week-old C57BL6 naïve females) and sex and age-matched mice at peak EAE after treatment with vehicle (PBS) or A13, respectively. Red blood cells were depleted prior to barcoding with the BD Rhapsody platform, library generation, and sequencing. The resulting data sets contain ECs, epithelial cells (type I and II), mesenchymal cells, monocytes, dendritic cells (DCs), neutrophils, T cells, B cells, and NK cells (Fig. 3a, b; Supplementary Fig. 9a). EAE and A13 treatment-induced gene expression changes in a variety of cell types including type I epithelial cells, matrix fibroblasts, and *Aplnr*+ ECs, according to the analysis of differentially expressed genes (DEGs) (Supplementary Fig. 9b). We initially focused on ECs to gain insight into the alterations in response to EAE and A13 treatment. By sub-clustering, we distinguished six endothelial subsets, which include arterial, venous, and lymphatic ECs together with several capillary EC subpopulations. *Apln* expression is prominent in Car4+ aerocytes[38-40], which have been previously associated with gas exchange in the lung. *Aplnr* is found in general capillary (gCap) ECs as well as in a smaller EC subpopulation expressing glutathione peroxidase 3 (*Gpx3*), an enzyme that protects against oxidative damage by catalyzing the reduction of hydrogen peroxide (Fig. 3c; Supplementary Fig. 10a, b). EAE caused a threefold increase in the relative fraction of lymphatic ECs, whereas the other EC subpopulations showed smaller changes (Supplementary Fig. 10a, b). gCap ECs were the most responsive endothelial population in terms of DEGs for both EAE and A13 treatment (Supplementary Fig. 10b). Gene Set Enrichment analysis using the Hallmark gene set[41] showed strong A13-induced reductions of TNFα/NFKB and transforming growth factor-β (TGFβ) signaling genes relative to vehicle-treated EAE lungs (Fig. 3d). Inflammation-associated genes, including *Cyp26b1* (cytochrome P450 family 26 subfamily B member 1), *Csf1* (colony-stimulating factor 1), *Sgms1* (sphingomyelin synthase 1), and *Zfp36* (zinc finger protein 36), were downregulated by A13 in EAE (Fig. 3d; Supplementary Fig. 10c).

We also analyzed immune cells in our lung scRNA-seq data (Supplementary Fig. 11a). EAE raises the relative abundance of neutrophils 6-fold, classical monocytes 2-fold and T-helper cells (Th) cells 4-fold, whereas B cells decrease by more than 60% (Fig. 3b). The abundance of Th cells as well as classical and non-classical monocytes was further elevated in EAE lungs treated with A13 (Fig. 3b, e, f; Supplementary Fig. 11a, b). The degree of myelin phagocytosis in MS coincides with CCR2+ monocyte invasion into the CNS[42]. Numerous mediators of inflammation, including C-C motif chemokine receptor 1 (*Ccr1*), matrix metalloproteinase-8 (*Mmp8*), S100 calcium-binding protein A9 (*S100a9*), and chemokine (C-X-C motif) ligand 2 (*Cxcl2*), were upregulated in classic monocytes in EAE lungs (Supplementary Fig. 11c, d). *Ccr1* transcripts and other inflammation-associated genes, such as *Fn1* (encoding fibronectin 1), were reduced after A13 treatment, but changes in gene expression relative to vehicle-treated EAE lungs were small (Supplementary Fig. 11c, d). We also looked into the changes affecting helper T cells (Th cells) and γδ T cells, which are critical players in the development of autoimmune diseases[43]. According to DEG analysis, EAE caused naïve T cells to

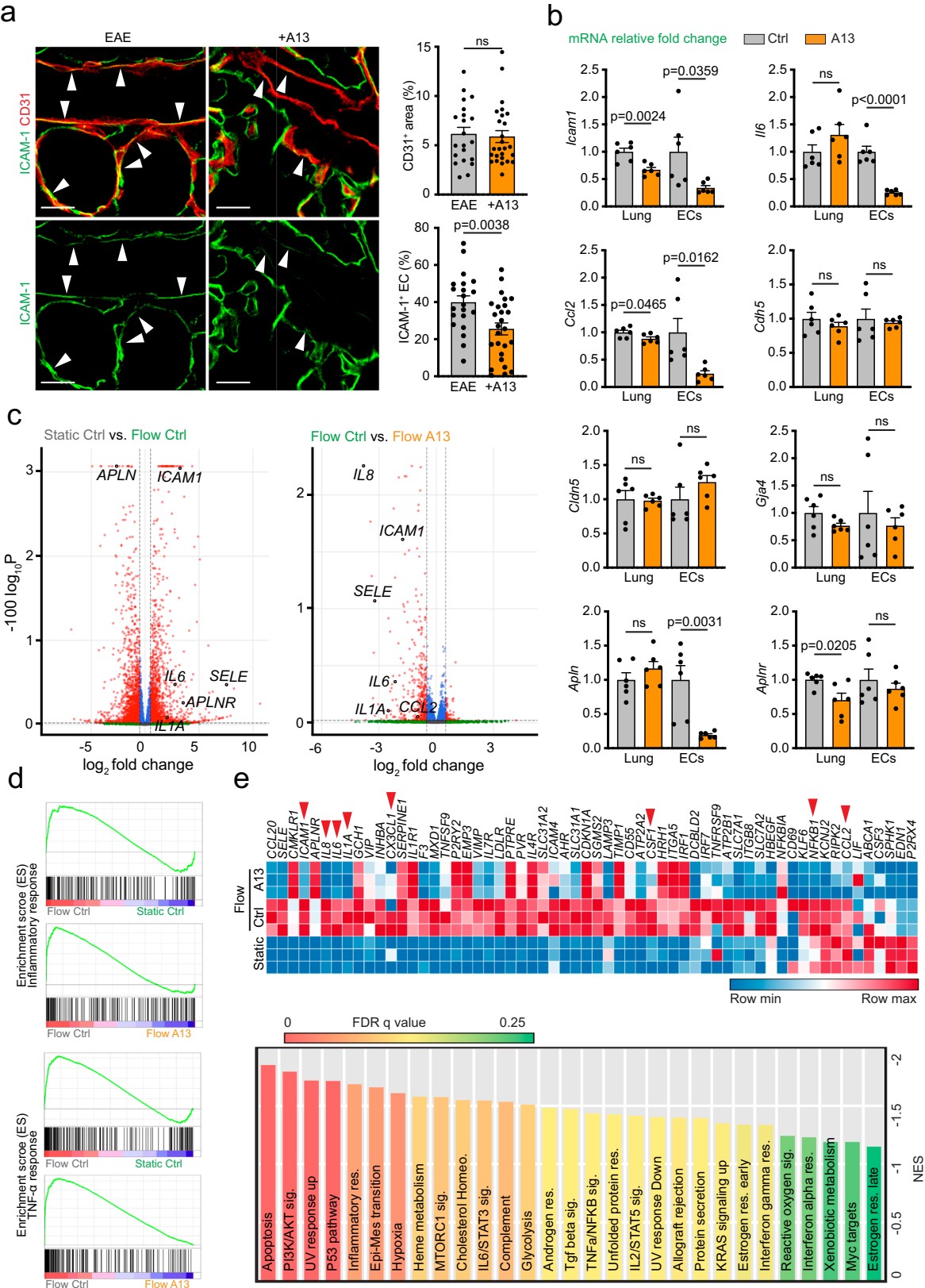

acquire T cell activation-related factors such as galectin 1 (*Lgals*), Thy-1 cell surface antigen (*Thy1*), the cell surface adhesion receptor CD44 (*Cd44*), and B-cell lymphoma 2 (*Bcl2*), a regulator of cell death (Supplementary Fig. 12a, b). Th cells also acquired expression of markers associated with T cell activation, including Ccr2 (*Ccr2*), galectin 1 (*Lgals*), and ADAM metallopeptidase domain 8 (*Adam8*)

(Supplementary Fig. 12a, c). Notably, administration of A13 had no or only small effects on the expression of these genes, indicating normal T cell representation and activation in A13-treated lungs. Finally, we compared gene expression in T cells from the lung and brain at peak EAE (Supplementary Fig. 13a, b). This revealed that T cell subpopulations isolated from the brain show much higher expression of

**Fig. 2 | A13 reduces inflammatory gene expression in ECs. a** Lung sections from A13-treated (+A13) or vehicle control EAE mice stained for CD31+ (red) ECs (arrowheads) and ICAM-1 (green). Note reduction of endothelial but not epithelial ICAM-1 by A13. Graphs on the right show quantitation of CD31+ area and ICAM-1+ area as a fraction of total EC area in randomly chosen areas. Scale bar, 20 μm. Error bars, s.e.m. Two-sided Student's *t*-test. *n* = 21 (EAE) and 26 (+A13) areas from 5 mice in each group. **b** qPCR of whole lung and sorted lung ECs after 7 days of A13 treatment. Data normalized to control (Ctrl). Error bars, s.e.m. Two-sided Student's *t*-test. *n* = 6 (2 technical replicates for 3 independent samples). **c** Volcano plots comparing gene expression differences in HUVECs with or without A13 treatment (Ctrl, A13) both under static conditions and flow (15 dyn/cm²). DEG analysis was performed based on two-sided Negative Binomial model. **d** Enrichment plots for inflammatory or TNF-α response genes after GSEA analysis for each comparison. The green curves represent the running sum of the weighted enrichment score, which is higher in the flow (15 dyn/cm²) control compared to the static control or A13-treated HUVEC. **e**, Heatmap of gene expression related to inflammation, comparing static and flow conditions (15 dyn/cm²) with/without A13 treatment (upper panel). GSEA Hallmark analysis showing enriched gene sets (FDR ≤ 0.25) in HUVEC treated with A13 under flow (lower panel).

activation markers[44–46] relative to lung. Some examples include interferon γ (*Ifng*), programmed cell death protein 1, also known as PD-1 or CD279 (*Pdcd1*), transforming growth factor β1 (*Tgfb1*), and chemokine ligand 5 (*Ccl5*) (Supplementary Fig. 13b). This result is consistent with the concept that T cells undergo multiple activation steps before exhibiting their full pathogenic potential inside the CNS. Indicating that the accumulation of immune cells in A13-treated lungs at peak EAE is not inducing pulmonary fibrosis, analysis of immunostained tissue sections revealed only a marginal elevation in collagen type I and immune cell cluster-containing areas of EAE showed no accumulation of α-smooth muscle actin (αSMA) in A13-treated samples relative to naïve lung (Supplementary Fig. 14a). At D30, the predominant expression of collagen type I and αSMA was observed in the bronchial and large arterial regions, reflecting coverage by smooth muscle cells, whereas only little signal was visible across the lung parenchyma, except for the few remaining immune cell clusters (Supplementary Fig. 14b).

### Immune cells in the lung form clusters near apelin receptor-positive gCap ECs

As mentioned above, immune cells accumulated in the lung before the onset of EAE symptoms and formed clusters, which were already visible at seven days after MOG35-55 immunization (Supplementary Fig. 1a). A13 treatment delayed this early formation of immune cell clusters, whereas the number of immune cells that were retained in the lung at EAE peak was strongly increased (Fig. 1f and Supplement Fig. 1a). These immune cell clusters resemble inducible bronchus-associated lymphoid tissue (iBALT), which are tertiary lymphoid structures generated by immune cell aggregation in non-lymphoid organs[47]. iBALT is frequently seen in the lungs of smokers and other patients with chronic pulmonary diseases, and is associated with the activation of the adaptive immune system[48,49]. Within these structures, which are often associated with bronchial epithelium, B cells are selected, T cells are primed, and both cell populations increase in number during inflammation[50,51]. CD45+ clusters, similar to iBALT structures, were detectable later in immunostained lung sections from A13-treated EAE mice compared to controls (Supplementary Fig. 1) and tended to be located further away from bronchi (Fig. 4a, b). Furthermore, A13 lungs contained higher proportions of T cells and less B lymphocytes compared to non-treated animals (Fig. 4c, d; Supplementary Fig. 15a, b). Relatively little is known about T cell migration within the pulmonary interstitial tissue, but previous work has proposed that the pulmonary vasculature is guiding this process[52]. Accordingly, we find that CD45+ immune cells were frequently detected near ECs in the alveolar microvasculature, which was enhanced by A13 treatment (Fig. 4e, f). T cell receptor (TCR) complexes are typically located at the rear end (uropod) of migrating T cells[53,54]. Analysis of CD3e distribution by immunostaining shows that the subunit of the TCR complex was more polarized in T cells of vehicle-treated EAE lungs compared to A13 samples (Fig. 4f, g). Taken together, these data suggest that A13 delays immune cell entry into the lung with implications for T cell polarization, the cellular composition of iBALT-like structures, and the resolution of immune cell cluster later during EAE.

Next, we investigated the spatial relationship between immune cells and APJ/Aplnr-expressing ECs in mice carrying the *Aplnr-CreERT2* allele together with *R26-mTmG* Cre reporter. *Aplnr-CreERT2*-mediated labeling of Aplnr+ gCap ECs showed that these cells are strongly associated with iBALT-like structures and scattered interstitial CD45+ immune cells in EAE mice (Supplementary Fig. 15c). Strikingly, more than 90% of CD45+ immune cells were already in contact with genetically labeled *Aplnr-GFP*+ ECs in naïve conditions (Supplementary Fig. 16a, b). Another model of tissue inflammation, LPS treatment, increased the fraction of CD45+ cells in the proximity of GFP-negative CD31+ ECs and CD31-negative structures, but the majority of immune cells (70%) remained associated with *Aplnr-GFP*+ ECs under these conditions (Supplementary Fig. 16a, b). Higher magnification images confirmed that *Aplnr-GFP*+ ECs were associated with CD45+ leukocytes, which is likely to indicate sites of leukocyte extravasation (Supplementary Fig. 16c). Conversely, a similar genetic strategy using *Apln-CreERT2*+ for the labeling of ECs shows that CD45+ cells were not preferentially associated with apelin+ (GFP+) aerocytes (Supplementary Fig. 16d).

### Regulation of leukocyte recruitment by apelin receptor

During our analysis of immune cell association with *Aplnr-CreERT2*-labelled cells, we observed that VE-cadherin immunostaining at junctions surrounding GFP+ ECs in EAE lungs was weaker relative to GFP-negative cells (Fig. 5a). This difference between GFP+ and unlabelled ECs was eliminated in A13-treated EAE lungs (Fig. 5a). To get further insight into the effect of apelin on EC behavior, we performed a series of in vitro experiments. As *APLNR* expression in HUVECs is low in the absence of flow, a receptor fusion construct carrying GFP at its carboxyterminus (APJ-GFP) was expressed by lentiviral infection and a similar strategy was used to generate control HUVECs expressing tdTomato fluorescent protein without APJ. In mixed cultures of tdTomato+ control and APJ-GFP+ HUVECs, VE-cadherin staining at junctions between APJ-GFP+ cells was less intense relative to junctions between tdTomato+ ECs, and this difference was reduced by A13 administration (Supplementary Fig. 17a). VE-cadherin function is controlled through the phosphorylation tyrosine residues in the cytoplasmic region of the cell adhesion molecule. While tyrosine 731 (Y731) in VE-cadherin is constitutively phosphorylated by Src family kinases (SFKs) and, in particular, the kinase Yes[55], dephosphorylation of Y731 is induced by interactions of ECs with T cells, which promotes T cell transendothelial migration[56]. Western blot analysis and immunostaining showed that A13 increased Y731 phosphorylation of VE-cadherin relative to vehicle control (Fig. 5b–d). The A13-induced increase in phospho-Y731 was abolished by treatment with PP1 (Fig. 5d), an inhibitor of all SFKs, consistent with previous reports linking Src family kinase activity to apelin[57,58].

Immunostaining also revealed localization of APJ-GFP at regions of cell-cell contact in vehicle control-treated cells, which was lost in A13-treated cells, resulting in perinuclear accumulation inside ECs (Fig. 5b; Supplementary Fig. 17a). To exclude the possibility that the C-terminal GFP fusion might alter the function of the GPCR, we also constructed an APJ-T2A-GFP lentiviral construct, in which self-cleavage generates separate APJ and GFP protein products. Coculture of APJ-

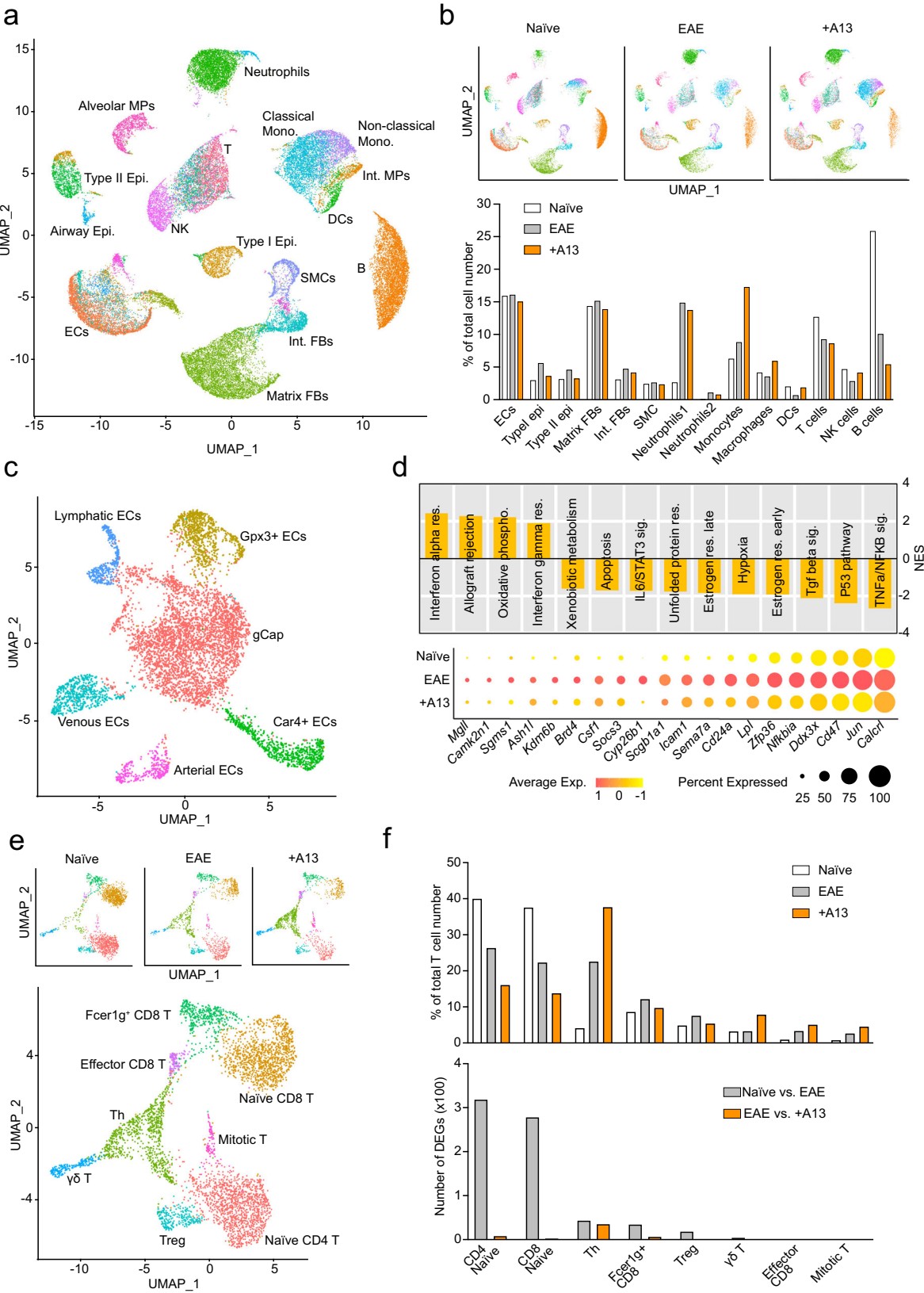

**Fig. 3 | A13-induced changes in EAE lungs. a** UMAP plot of total cells from lung with color-coded cell types. Single cells were dissociated from three mouse lungs in each group (Naïve, EAE, EAE + A13) at D16 without any depletion, except red blood cell lysis. scRNA-seq was performed using BD Rhapsody and Illumina NextSeq 500 platforms. **b** UMAP plots of individual samples and bar graphs showing the population ratio of each cell type. **c** UMAP plot of the EC subset. **d** GSEA Hallmark analysis of enriched gene sets (FDR ≤ 0.25) in gCap ECs, comparing +A13 vs. EAE.

Dot plot shows the expression and percent expression of inflammatory-related genes in gCap ECs for each condition. **e** UMAP plots of T cell subsets. **f** Population ratio of each T cell subset in each condition (upper panel) and a number of differentially expressed genes (DEGs) (adjusted *p*-value < 0.05, log2 fold-change>0.25) comparing Naïve vs. EAE or +A13 vs. EAE. DEG analysis was performed based on a two-sided Negative Binomial model.

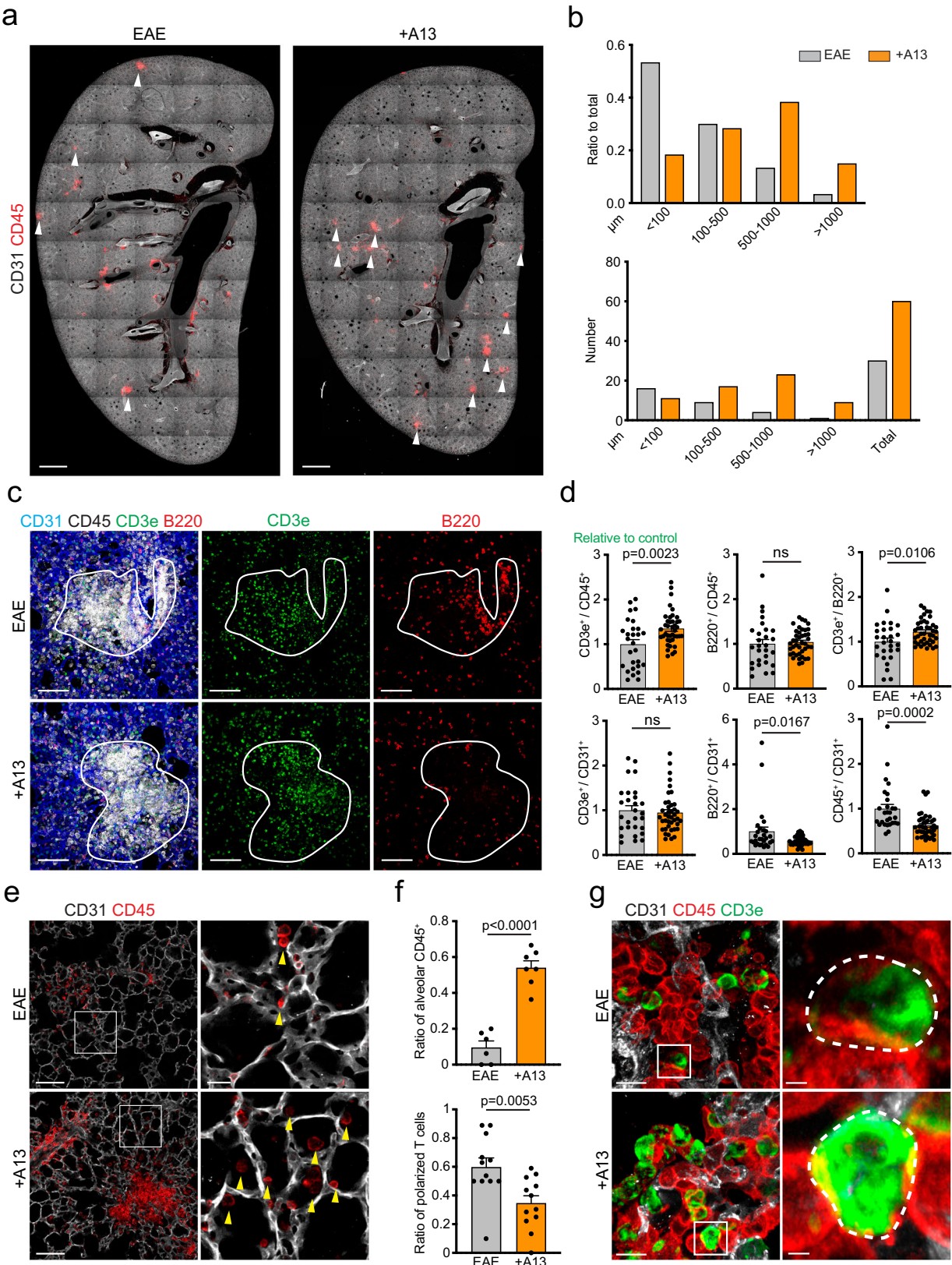

T2A-GFP and tdTomato HUVECs confirmed weaker VE-cadherin staining at junctions between GFP+ cells relative to tdTomato+ cells, and this difference was diminished by A13 treatment (Supplementary Fig. 17b). As no major changes in *Cdh5* transcript level were observable in pulmonary EC subpopulations in vivo or in HUVECs in vitro, the effects of A13 on VE-cadherin are likely to be posttranscriptional (Supplementary Fig. 17c, d).

ICAM-1, a known mediator of leukocyte arrest, intravascular crawling and transendothelial migration, is upregulated in EAE[12,20,33], but was reduced by A13 treatment in pulmonary ECs in vivo and in

**Fig. 4 | A13 alters the formation of immune cell clusters in EAE lung.**
**a** Immunostaining of CD45+ (red) immune cell clusters (arrowheads) and CD31 (white) in lung sections at D11 of EAE in vehicle-treated (EAE) and A13-injected mice (+A13). Scale bar, 1 mm. **b** Quantification of the distance between immune cell clusters and nearest bronchus in lung sections (*n* = 5 mice). Lower panel shows the number of immune cell clusters per distance and upper panel shows the ratio of clusters per distance relative to all clusters for each condition. **c** Immunostaining of EAE and +A13 lung sections at D11 (EAE onset) for CD31 (blue), CD45 (white), CD3e (green), and B220 (red). White lines outline immune cell clusters. Note low abundance of B220+ B cells in +A13 samples. Scale bar, 100 μm. **d** Quantitation of CD3e+ or B220+ area normalized to CD45+ or CD31+ area in EAE and +A13 immune cell clusters (as shown in **c**). Error bars, s.e.m. Two-sided Student's *t*-test. *n* = 28 (EAE) and 41 (+A13) areas from 5 mice for each condition. **e** Representative images of lung sections stained for CD31 (white) and CD45 (red) for EAE and +A13 at D16 (peak EAE). Yellow arrowheads indicate immune cells located in alveolar area. Scale bar, 100 μm (left) and 20μm (right). **f** Quantitation of CD45+ cells per alveolar area (as shown in **e** and ratio of T cells with polarized CD3e staining (green, as shown in **g**). Error bars, s.e.m. Two-sided Student's *t*-test. *n* = 6 mice (EAE) and 7 mice (+A13) for **e**, *n* = 12 each for **g**. Scale bar, 5μm (left) and 1μm (right). **g** Representative overview images and higher magnifications of insets showing CD3e immunostaining (green) with dashed lines marking outline of cells based on CD45 signal (red).

HUVECs in vitro (Fig. 2a-c). We therefore employed in vitro assays to directly assess the effects of A13 on T cell adhesion and transmigration across HUVECs and on ICAM-1 expression. Analysis of transmigration under flow confirmed that APJ-GFP expression by HUVECs increased T cell adhesion relative to tdTomato+ control HUVECs. A13 reversed this effect leading to low levels of T cell adhesion to APJ-GFP+ cells and had no significant effect on tdTomato+ control HUVECs (Fig. 5f, g). Live imaging analysis showed that crawling distances of T cells on HUVECs were slightly increased by A13 administration (Fig. 5g). The same approach also indicated that A13 treatment of APJ-GFP+ cells led to a small but significant increase in transcellular migration events relative to paracellular (junction-mediated) migration (Fig. 5g, h). In transwell assays, the migration of T cells across an APJ-GFP HUVEC monolayer in response to CXCL12/SDF-1 was also reduced by A13 (Fig. 5i).

Together, these results show that apelin reduces the transendothelial migration of T cells, but possibly also of other immune cells, which involves reduced adhesion to ECs.

## Ablation of apelin receptor in ECs recapitulates the effect of A13 in EAE

Ligand binding induces signaling by G protein-coupled receptors but, depending on ligand concentration and other factors, it can also limit signal transduction by triggering GPCR internalization and removal from the cell surface in a process termed desensitization[59]. Indeed, it was shown that A13 can desensitize apelin receptor, which involves internalization via clathrin-coated vesicles[60]. As our own in vitro experiments supported the possibility of A13-induced APJ/Aplnr internalization (Fig. 5b), we generated Cy5-tagged fluorescent version of A13 (A13-Cy5) for stimulation experiments. At 2 hours after treatment, internalized A13-Cy5 speckles were visible inside APJ-GFP-expressing HUVECs but not in tdTomato+ control cells. In vehicle-treated HUVECs, APJ-GFP was readily detectable at cell-cell contact sites and this signal was lost after A13-Cy5 administration and APJ internalization (Supplementary Fig. 17e). The sum of the data above indicates that A13 removes its receptor from the cell surface through internalization, resulting in a transient loss-of-function situation limiting inflammatory leukocyte transmigration across the endothelial monolayer.

Next, we performed bulk RNA-sequencing of cultured HUVECs in a loss-of-function setting, namely after siRNA-mediated knockdown of apelin receptor expression (*siAPLNR*). This approach led to the downregulation of inflammatory and TGFβ signaling-related gene expression both under fluid shear stress and static conditions (Fig. 6a, b). Similar to A13 treatment (Supplementary Fig. 6), *siAPLNR* reduced the low baseline expression of *ICAM1*, *VCAM* and *SELE*, but the effects were not as pronounced as those measured after A13 administration (Supplementary Fig. 18a). Moreover, *IL8*, *IL6* and *CCL2*, which were downregulated by A13 administration, were increased in *siAPLNR* HUVECs (Supplementary Fig. 18a).

For functional studies in vivo, we generated EC-specific *Aplnr* knockouts (*Aplnr*iECKO) by interbreeding of *Cdh5-CreERT2* mice[61] with animals carrying a loxP-flanked *Aplnr* gene[62] and tamoxifen treatment. In contrast to brain, both the endothelium and the epithelium in the lung shows constitutive expression of ICAM-1, which is further upregulated in response to inflammation[63,64]. Staining of lung sections confirmed constitutive ICAM-1 expression in control littermates, whereas ICAM-1 immunofluorescence was strongly reduced in *Aplnr*iECKO mutants (Supplementary Fig. 18b). Decrease of ICAM-1 was also seen in pulmonary ECs of adult (8-week-old) *Aplnr*iECKO mutants (treated with tamoxifen 1 week earlier), without alterations in the density of pulmonary vessels (Fig. 6c, d). *Aplnr*iECKO mutants showed reduced accumulation of CD45+ cells in response to LPS exposure (Supplementary Fig. 19a, b). Importantly, EAE progression was delayed and disease severity reduced in *Aplnr*iECKO mutants relative littermate controls (Fig. 6e). Flow cytometry showed decreased immune cell infiltration into *Aplnr*iECKO brains and elevated accumulation of immune cells in the lung compared with control littermates (Fig. 6f, g). Thus, genetic inactivation of apelin receptor expression in ECs recapitulates critical aspects of the A13-induced protective phenotype, arguing that decreased apelin receptor function is a common feature in both conditions. The sum of these data argues that *Aplnr*+ gCap ECs facilitate immune cell entry into the lung in the EAE model but also in response to LPS treatment. Immune cell recruitment into the lung is impaired by A13, which induces internalization of apelin receptor, or by genetic inactivation of the *Aplnr* gene in ECs.

## A13 treatment during EAE development ameliorates disease symptoms

To investigate the therapeutic potential of A13 in EAE and also rule out potential effects of the peptide on the initial immune response after MOG35-55 administration, mice were treated with A13 every second day from D7 or daily from D11 (at disease onset) (Fig. 7a). Strikingly, both late treatment regimens significantly reduced disease severity relative to vehicle control animals (Fig. 7a). A13 treatment from D7 also delayed the onset of disease symptoms. Consistent with our previous results, lung lobe sections of mice treated with A13 from D11 presented a higher number of CD45+ immune clusters relative to vehicle control at D20 (Fig. 7b, c). Moreover, the cerebellum of A13 mice at D20 showed reduced immune cell infiltration together with lower ICAM-1 expression (Fig. 7b, d). These results support that the benefits of A13 are not caused by the suppression of the initial immune response in the EAE model (before D7 or D11, respectively), but they also raise the possibility that A13 might allow therapeutic intervention in an early stage of disease development with potential relevance for MS.

It has been previously suggested that A13 might be a disease biomarker that is elevated in the serum of MS patients relative to healthy controls[65]. We, therefore, analyzed serum samples of healthy controls and MS patients in relapse or remission (Supplementary Fig. 20a). This revealed significantly higher levels of serum A13 in healthy males relative to healthy women. Female MS patients exhibited an increase in serum A13 during relapse and a smaller, statistically not significant elevation during remission, which raised levels to those seen in healthy males. Disease status had no measurable impact in male samples (Supplementary Fig. 20a). Given that MS is more prevalent in women than men[66], these findings raise the interesting possibility that higher levels of A13 in male serum might be potentially relevant for protection against the disease in humans.

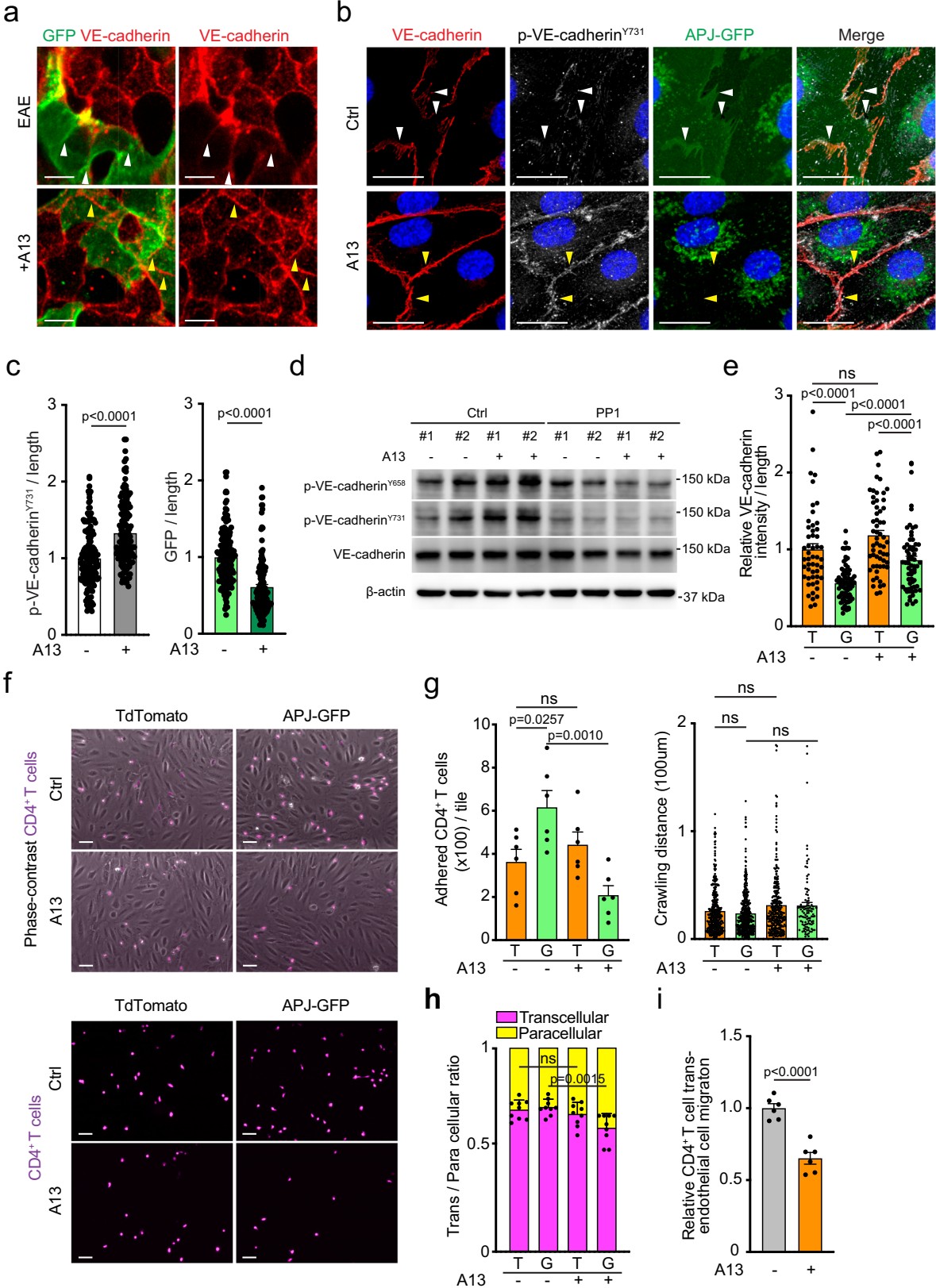

## Discussion

The etiology of MS in patients remains incompletely understood, but research using EAE as an animal model has indicated a role of T cell reprogramming in the lung, which induces a migratory phenotype and enables the trafficking of these cells into the CNS[8,18]. Within the lung, chemokine signals direct T cells into bronchus-associated lymphoid tissue before they re-enter the blood circulation through lung-draining lymph nodes[8]. These findings are consistent with other reports showing that iBALT in the lung can mediate substantial respiratory immune responses even without an essential role of traditional secondary lymphoid organs[47,67]. Our study shows that A13 delayed immune cell recruitment to the lung but also the subsequent

**Fig. 5 | A13 increases junctional VE-cadherin and reduces immune cell trafficking. a** High magnification confocal image of immune cell cluster in tamoxifen-treated *Aplnr-CreERT2 R26-mTmG* lung stained for VE-cadherin (red). White arrowheads mark VE-cadherin+ junction of APJ+ ECs. Scale bar, 50 μm. **b** Confocal image of HUVEC expressing apelin receptor with c-terminal GFP-tag (APJ-GFP) with or without A13 treatment, stained for VE-cadherin (red), phospho-VE-cadherin$^{Y731}$ (white), APJ-GFP, and DAPI (blue). Note weak phospho-Y731 signal surrounding APJ-GFP+ cells (white arrowheads) but increase in response to A13 (yellow arrowheads), which also reduces APJ-GFP at cell-cell borders. Scale bar, 20 μm. **c** Quantification of phospho-VE-cadherin$^{Y731}$ and APJ-GFP signal at cell perimeter, as indicated. Error bars, s.e.m. Two-sided Student's *t*-test. *n* = 157 (vehicle treated), *n* = 153 (A13 treated) for both phospho-VE-cadherin$^{Y731}$ and APJ-GFP. **d** Western blots of phospho-VE-cadherin$^{Y658}$, phospho-VE-cadherin$^{Y731}$, total VE-cadherin, and β-actin (loading control). APJ-GFP expressing HUVEC were treated with A13 and/or PP1, as indicated. Results from two separate stimulation experiments (#1, #2) are shown. **e** Quantitation of VE-cadherin at cell-cell contacts in tdTomato+ control HUVECs (T) and APJ-GFP+ HUVECs (G) (both pre-treated with LPS, with or without A13

treatment, as indicated). Error bars, s.e.m. Two-sided Student's *t*-test. *n* = 51 for tdTomato+ HUVECs and *n* = 75 for APJ-GFP+ HUVECs treated with vehicle; *n* = 58 for tdTomato+ HUVECs and *n* = 64 for APJ-GFP+ HUVECs treated with A13. **f** Merged images showing phase-contrast of HUVECs (pretreated with LPS) and fluorescently labelled CD4+ T cells (top). Bottom panels show isolated fluorescent channel highlighting A13-induced reduction of adherent CD4 + T cells (purple). Representative movies of brightfield channels are provided as Supplementary Movies 1-4. Quantitation of adherent CD4+ T cells, crawling distance, and ratio of transcellular vs. paracellular trans-endothelial migration on tdTomato+ control HUVECs (T) and APJ-GFP HUVECs (G) with or without A13 treatment, as indicated. Error bars, s.e.m. Two-sided Student's *t*-test. Scale bar, 50 μm. *n* = 6 areas for each group from 2 independent experiments (**g**, left panel). *n* = 244 cells for all conditions except APJ-GFP+ HUVECs +A13 (*n* = 104) (**g**, right panel). *n* = 9 areas for each group from 2 independent experiments (**h**). **i** Quantification of CD4+ T cell migration across APJ-GFP HUVECs in transwell assay. 6 individual wells/condition from 2 independent experiments. Error bars, s.e.m. Two-sided Student's *t*-test.

resolution of immune cell clusters in the pulmonary parenchyma, which led to reduction of the number of immune cells in peripheral blood and in mediastinal lymph nodes. These changes were associated with reduced entry of pathogenic T cells into the CNS, resulting in delayed EAE progression and milder disease symptoms relative to vehicle-treated EAE animals. We also provide evidence that treatment at the onset of EAE symptoms can reduce disease score, which suggests that further exploration of the therapeutic potential of A13 could be worthwhile. At the same time, limitations of EAE as a transient model of MS need to be considered. EAE either leads to death or partial recovery of the affected animals[68,69]. The latter is associated with the deletion of encephalitogenic T cells and, in models involving active immunization, resistance to subsequent induction of EAE. Therefore, the role of A13 in chronic neuroinflammation cannot be studied. The primary immune cell types implicated in the manifestation of symptoms are also not identical for the human disease and its animal model. CD8+ T-cells and B-lymphocytes have been associated with progressing inflammation and tissue damage in MS, whereas pathogenesis in EAE is mediated by auto-reactive CD4+ T-cells[70].

The effect of A13 is not restricted to EAE and the ligand also successfully reduced pulmonary inflammation in response to intranasally administered LPS, which is consistent with various previous reports showing that A13 can protect against acute lung injury[29,31,71,72]. Most of these studies have attributed the beneficial effects of apelin to altered signaling in immune cells[29,31,72], whereas the EC-specific knockout of *Aplnr* and in vitro experiments in our study indicate an important and direct role of ECs through the regulation of leukocyte transendothelial migration. In our in vivo experiments, we noted that the protection against EAE symptoms is more efficient in A13-treated mice than after EC-specific *Aplnr* inactivation. One possible explanation could be effects on other cell populations even though *Aplnr* expression is largely, but not completely, confined to ECs. Moreover, A13 might also trigger APJ signaling and not only desensitization of the GPCR, whereas the genetic approach generates an irreversible loss of apelin receptor function without active signal transduction. Finally, it should also be considered that systemic administration of A13 can exert vasodilatory and anti-hypertensive effects, which has been, at least in part, attributed to the modulation of vascular smooth cell function[73,74].

Our data on serum levels of A13 in MS patients support its use as a biomarker at least in female patients. Two previous studies had reported variable results between male[57] and female MS patients[75], but it has to be considered that sample numbers were small in both studies and that disease state, comorbidities and treatment differences might result in substantial variability. Given that MS is more prevalent in women than men[66], these findings raise the interesting possibility that higher levels of apelin in male serum might be potentially relevant for protection against the disease. However, it also needs to be considered that the biological effect of constant exposure to endogenous apelin

might not be equivalent to acute A13 administration. The functional modulation of *Aplnr*+ ECs, which are, as our data show, associated with leukocyte transendothelial migration in the lung, by apelin and other factors deserves further investigation in future studies.

At the mechanistic level, we link the biological effect of A13 to reduced T cell adhesion to ECs (Supplementary Fig. 20b). E-selectin, one of the critical TEM mediators in peripheral tissues that is down-regulated by A13, functions in the rolling interactions that decelerate leukocytes and position them in close proximity to the endothelial monolayer[76]. A13 also reduces the endothelial expression of ICAM-1 and VCAM-1, which mediate leukocyte adhesion through interactions with integrin receptors and are both upregulated by inflammation. Interestingly, previous work has shown that effector T cells are temporarily entrapped in the pulmonary vasculature on their way to lymph nodes during systemic inflammation[63]. T cell entrapment in the lung is reduced in mice lacking ICAM-1 and the related ICAM-2 molecule, whereas neutrophil recruitment is unaffected[63]. Endothelial VCAM-1 participates in leukocyte diapedesis, which also involves downstream signalling events that help to open endothelial junctional cell-cell contacts[77–79]. This might also explain why some of our findings link A13 and APJ/Aplnr internalization to junctional localization of VE-cadherin, whereas *Cdh5* transcript levels are not altered. Other processes, including altered expression of the inflammatory cytokines CCL2 and IL6 (Supplementary Fig. 20b), might also help to explain why A13 administration delays immune cell trafficking through the lung and thereby suppresses the development of EAE. Similarly, the persistence of immune cell clusters in the lung of A13-treated mice at peak EAE might be caused by altered cytokine expression, the initial delay in immune cell accumulation in the lung or impaired migration towards lymphatic vessels. Regarding the latter, it should be noted that various studies have revealed important roles of apelin signaling in the growth and remodeling of the lymphatic vasculature[80–82]. Genetic approaches could be used to distinguish between the effects of A13 on the ECs of blood vessels or those of the lymphatic vasculature in future studies.

There is currently no cure for autoimmune disorders such as MS, and available treatments can have harmful side effects and need to be carefully monitored because they generate an increased risk of infection[83,84]. Further research will be needed to address whether A13 or other mediators of apelin signaling might be useful for the suppression of acute or chronic inflammation either alone or in combination with other treatment options.

## Methods

### Mice and EAE mouse model

To generate EC-specific *Aplnr* knockouts (*Aplnr*$^{iECKO}$), mice carrying a loxP-flanked *Aplnr* allele[62] were interbred with *Cdh5-CreERT2*[61] transgenic animals. For analysis at postnatal day 6 (P6), 50 μl of 1 mg/ml tamoxifen (Sigma T5648) was administered at P1, P2, and P3. For

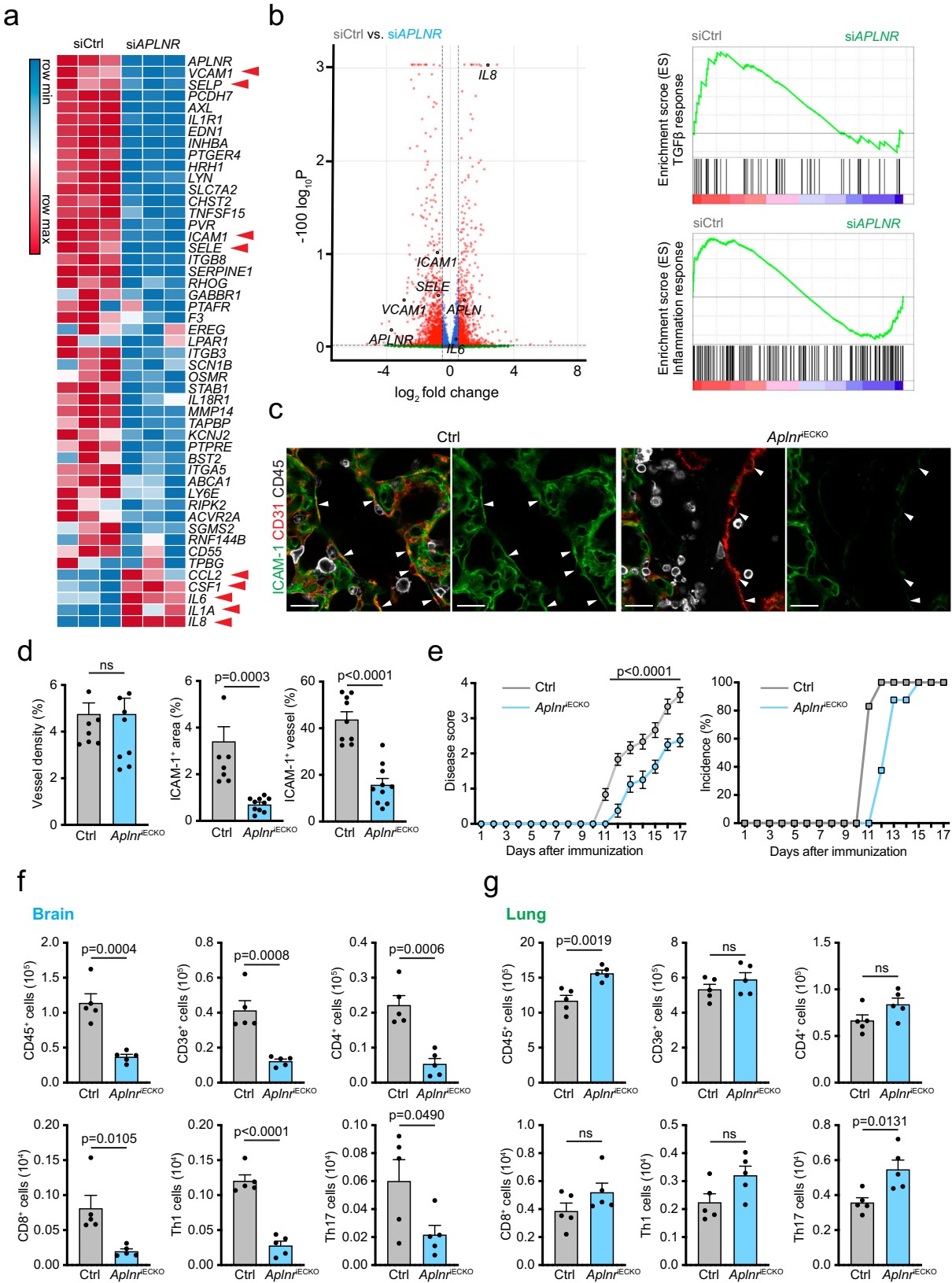

studies in adult mice, 200 µl of 10 mg/ml tamoxifen were given on 3 consecutive days, 1-2 weeks prior to analysis or EAE induction. For the analysis of interactions between ECs and immune cells, *Aplnr-CreERT2*[85] or *Apln-CreERT2* mice[86] were bred to *R26-mTmG* Cre reporter animals[87]. The resulting offspring were treated with tamoxifen as described above. EAE was induced in 8-12-week-old C57Bl/6 wild-type

females or in *Aplnr*[iECKO] or *Aplnr-CreERT2 R26-mTmG* females and littermate controls, respectively. Experimental procedures adhered to previous studies[88]. Briefly, a subcutaneous injection of 150 µl of MOG$_{35-55}$ (112 µg/ml in emulsion with Complete freunds adjuvant (CFA)) was administered near the tail base, along with 100 µl of Pertussis toxin (Sigma P7208, 2 µg/ml) via the tail vein. The following day, 100 µl of

**Fig. 6 | Apelin receptor ablation in ECs. a** Heatmap showing expression of inflammation-related genes in HUVECs treated with control siRNA (siCtrl) or *APLNR* siRNA (si*APLNR*) under flow conditions. Red arrowheads indicate genes mentioned in text. **b** Volcano plot comparing gene expression of siCtrl and si*APLNR* treated HUVECs under flow (left panel). Plots show enrichment of TGF-β or inflammatory response genes in GSEA analysis (right panels). DEG analysis was performed based on two-sided Negative Binomial model. **c** Confocal images of lung sections from EC-specific *Aplnr* knockout mice (*Aplnr*[iECKO]) and littermate controls at EAE peak stained for ICAM-1 (green), CD31 (red) and CD45 (white). Arrowheads indicate

endothelial ICAM-1 and downregulation in *Aplnr*[iECKO] lung. Scale bar, 20 μm. **d** Quantitation of vessel density (CD31+ area per unit area) and ratio of ICAM-1+ (green) area per total area or CD31+ area. Error bars, s.e.m. Two-sided Student's *t*-test. *n* = 7 for Ctrl, *n* = 10 for Aplnr[iECKO]. **e** Disease score and incidence rate of EAE in 8-12 week-old female *Aplnr*[iECKO] mice and age and sex-matched littermate controls (Ctrl). Error bars, s.e.m. two-way ANOVA. *n* = 6 mice for each condition. Number of CD45+ cells, CD3e+ T cells, CD4+ T cells, CD8+ T cells, Th1 cells, and Th17 cells in Ctrl and *Aplnr*[iECKO] whole brain (**f**) and lung (**g**) at peak EAE. Error bars, s.e.m. Two-sided Student's *t*-test. *n* = 5 mice/group.

---

Pertussis toxin (2 μg/ml) was administered once again. Throughout the experiment, 100 μl of A13 (10 nM, Bio-techne, #2420) or PBS was administered intraperitoneally to the mice, and body weight and disease severity according to the EAE clinical severity scale[89] were assessed daily. During EAE experiments, vehicle control and A13-treated mice were kept in separate cages within the same room. Animal experiments were not blinded because treatments had to be indicated on cage labels and in our animal database for legal reasons. The size of experimental groups was selected on the basis of previous experiments[88]. Moreover, A13-treated animals showed obvious phenotypic differences during the course of the EAE experiments. Mice were euthanized and dissected on the designated day as detailed in the main text, following terminal anesthesia. All animals were housed in a dedicated pathogen-free facility, and experiments were conducted in compliance with applicable laws and institutional guidelines. The experiments were performed after ethical review and with the necessary permissions granted by the Landesamt für Natur, Umwelt und Verbraucherschutz (LANUV) of North Rhine-Westphalia, Germany.

### Intranasal LPS challenge
A concentration of 2.5 mg/ml of LPS (Sigma, L2630) was dissolved in PBS, with or without A13 (10 nM). 8-12 week-old female C57Bl/6 wild-type mice were sedated using isoflurane. Subsequently, 20 μl of LPS solution was administered to the nostrils of each mouse under anesthesia. Mice were kept in an upright position until they regained consciousness. Animals were sacrificed and lungs were prepared for further processing after 18 h for section staining and 4 or 24 h for BALF analysis.

### Organ isolation and immunostaining
Mice were examined for reflexes following anesthesia induced by Rompun and Ketamine. Anesthetized mice had their caudal vein cut and the chest was opened. DPBS (Dulbecco's Phosphate-Buffered Saline) was injected through the right ventricle until the lung became pale. Subsequently, 0.1% low melting agarose was administered via the trachea, and the trachea was clamped for 15 minutes. The organs were then prepared and fixed in 4% PFA (Paraformaldehyde) overnight at 4 °C. Vibratome (Leica VT1200S) was used to cut the organs into 100 μm or 200 μm slices. Sections were stored in an anti-freezing media at −20 °C until use. To remove the anti-freezing media prior to staining, sections were rinsed three times with PBS (Phosphate-Buffered Saline). A blocking buffer and antibody diluent containing 5% Donkey serum in 0.1% Triton X-100 PBST (Phosphate-Buffered Saline with 0.1% Tween-20) were used. The blocking and antibody treatment was performed overnight at 4 °C. Finally, samples were washed three times for 15 minutes each with 0.1% Triton X-100 PBST at room temperature.

### Analysis of CD3e polarity
Utilizing Fiji, individual cells identified through CD45 staining were delineated and a macro was employed to partition these cell into semicircles, measuring CD3e staining intensity on each side separately. The ratio of intensity between one semicircle and the other was computed. Subsequently, the dividing line oft he semicircle was rotated by 45 degrees and intensity measurement and calculation was

repeated three more times. The results from these four calculations were aggregated for each cell and cells with calculated values exceeding 2 or below 0.5 were classified as polarized.

### Antibodies
A list of antibodies used in the study is shown in Table 1.

### Lung dissociation for the scRNA sequencing and EC isolation
Lung tissue was dissociated using a solution containing papain (Worthington, LK003150) and liberase (Roche, 5401054001). Dissociation was carried out for 40 minutes at 37 °C in a water shaker. Cells were collected by centrifugation and the pellet was resuspended in MEM. The cell suspension was then filtered through a 100 μm mesh to remove large debris, followed by treatment with RBC lysis buffer (Sigma, R7757) to eliminate red blood cells. A second filtration step using a 40-μm mesh was performed. The cell count was determined using a cell counter (Logos, L10001). For single-cell RNA sequencing, the cell suspension was diluted and aliquoted at a concentration of 40,000 cells per 100 μl in FACS buffer (0.2% FBS, 2 mM EDTA in PBS). The BD Rhapsody single-cell analysis system was utilized to prepare the single-cell RNA libraries, which were subsequently sequenced using the Illumina Next-seq 500 platform with a mid-output setting. To isolate ECs, CD45+ cells were depleted using the MACS method and CD45 microbeads (Miltenyi Biotec, 130-052-301). Next, cells were stained with PDPN-PE (12-5381, eBioscience), CD45-Pecy7 (25-0451-82, eBioscience), and CD31-APC (FAB3628A, R&D) for 30 minutes at 4 °C. The stained cells were washed three times with FACS buffer. Using FACS ARIA (BD, FACSARIA IIIU), CD45-negative cells were gated, and CD31 + PDPN-cells were sorted for further applications.

### FACS analysis for the brain and lung-infiltrating immune cells
Mice were anesthetized using Ketamine and Rompun. Peripheral blood was collected from the caudal vein using a needle and syringe into an anticoagulant tube. To remove red blood cells, the collected blood was treated three times with RBC lysis buffer (Sigma cat. R7757). Cells were then kept in FACS buffer on ice for further FACS analysis. For the analysis of brain and lung tissue, mice were perfused with DPBS as described in the method section on organ isolation. Lungs and brains were minced in a 2 ml eppendorf tube, and 1 ml of FACS buffer was added until the color became homogeneous. Lung and brain pieces were dissociated in FACS buffer in a 6-well plate using a 100 μm mesh and plunger. The cell suspension was collected with 10 ml of FACS buffer and centrifuged to obtain a cell pellet. Cell pellets were resuspended in 37% Percoll and loaded on top of 70% Percoll (Cytiva, 17089102). After centrifugation, the mononuclear cell layer was collected. Cell pellets were resuspended in 1 ml of FACS buffer and the cell concentration was measured using a cell counter. For CD45, CD4, and CD8 T cell analysis, $10^5$ cells were stained directly with CD45-APCcy7, CD3e-FITC, CD4-APC, and CD8-PE at 4 °C for 20 minutes, followed by three washes with FACS buffer. The stained cells were then analyzed using a FACS analyzer (FACSymphony A3). The remaining cells were cultured with 10% FBS RPMI 1640 medium supplemented with 10 ng/ml PMA (Sigma, P1585), 1 μg/ml ionomycin (Sigma, IO634), and 10 μg/ml Brefeldin A

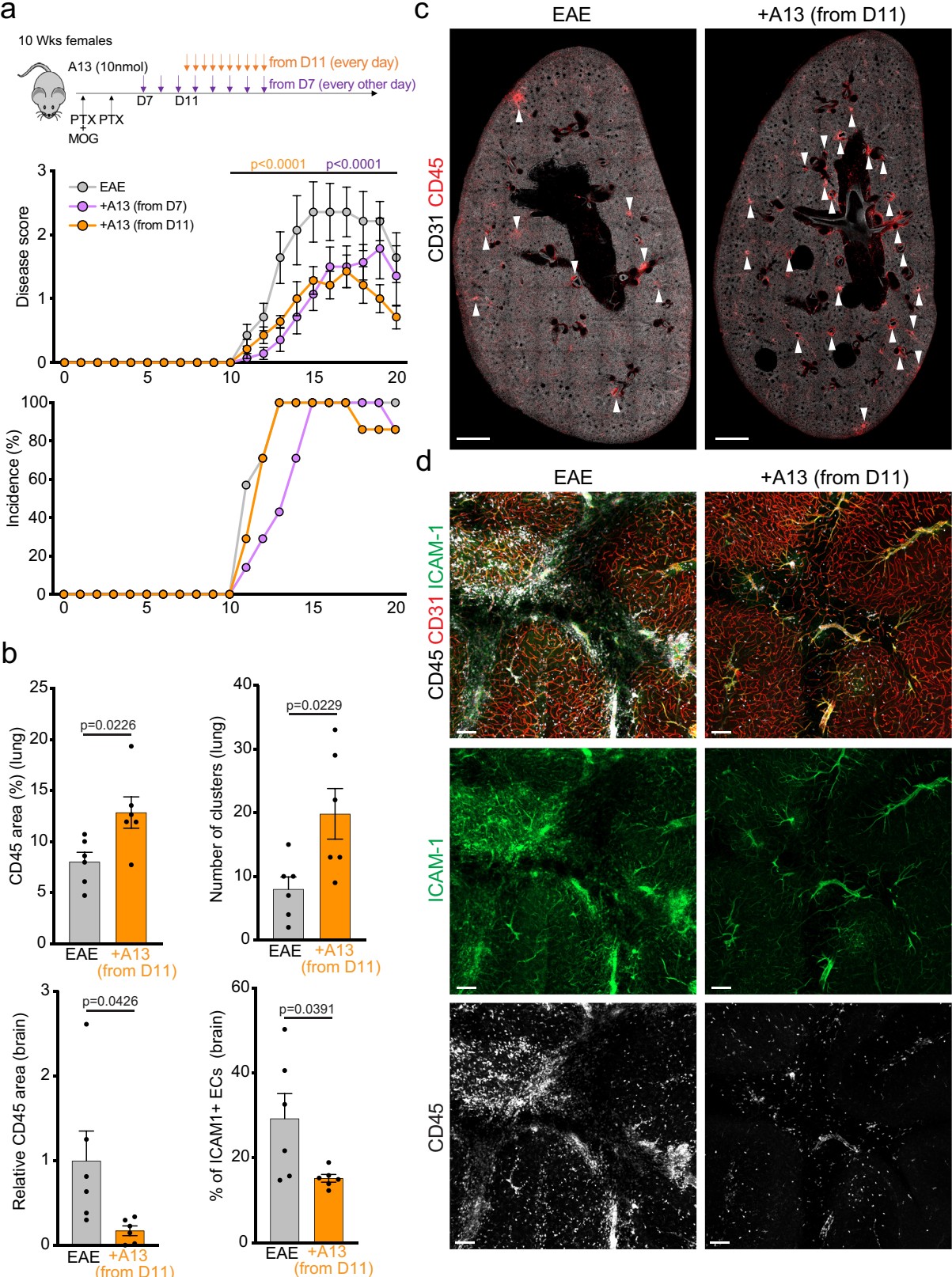

(Sigma, B6542) for 6 hours. For IFNγ-PECy7 and IL-17A Alexa Fluor 488 staining, the BD Cytofix/Cytoperm kit (BD, VDV554715) was used according to the manufacturer's instructions, followed by staining with CD45-PE and CD4-APC. The stained cells were then analyzed using a FACS analyzer (FACSymphony A3). Gating strategies are shown in Supplementary Figs. 21 and 22.

**Cell culture and flow experiment**

HUVECs were cultured in EC proliferation medium (Provitro, 201 0001) on dishes coated with 0.5% gelatin. Cells were used in experiments until passage 5. For the flow experiment, $10^5$ cells were seeded on gelatin-coated 0.6 mm u-slide I Luer chambers (ibidi 80186). The media was changed 4 hours after seeding to complete media before

**Fig. 7 | Therapeutic administration of A13 in the course of EAE. a** Time course of $MOG_{35-55}$ immunization and A13 administration and analysis. Female C57Bl/6 mice aged 8-12 weeks received i.p. A13 or vehicle injections every other day from D7 or daily from D11 (Onset). Shown are incidence rate and disease scores on the indicated days after immunization. Error bars, s.e.m. two-way ANOVA, $n = 7$ mice per group. **b** Quantification of immune cells infiltrating the brain and lung per area, number of clusters in the lung and ICAM-1 + EC ratio in the brain at D20 of EAE and +A13 from D11 onwards ($n = 6$ mice per group). Error bars, s.e.m. Two-sided Student's $t$-test. **c** Overview confocal images of EAE or +A13 (D11 onwards) lung lobe sections from D20 immunostained for CD31 (white) and CD45 (red). Scale bar, 1 mm. White arrowheads indicate immune cell clusters. **d** Representative confocal images of control (EAE) and +A13 (D11 onwards) cerebellum at D20 immunostained for CD45 (white), ICAM-1 (green) and CD31+ (red). Scale bar, 100 µm. Disease score, 2 for EAE and 1 for +A13.

## Table 1 | List of antibodies used in the study

| Antibody | Company | Catalog number | Usage | Dilutions |
|---|---|---|---|---|
| alpha-SMA-eFluor-660 | eBioscience | 50-9760-82 | Immunostaining | 1:400 |
| anti chicken Alexa Fluor-488 | Jackson Laboratories | 703-545-155 | Immunostaining | 1:400 |
| anti goat Alexa Fluor-647 | Invitrogen | A21447 | Immunostaining | 1:400 |
| anti goat Alexa-594 | Invitrogen | A11058 | Immunostaining | 1:400 |
| anti goat DyLight-405 | Jackson Immuno Research | 705-475-147 | Immunostaining | 1:400 |
| anti hamster Alexa Fluor-647 | Invitrogen | A21451 | Immunostaining | 1:400 |
| anti mouse Alexa Fluor-405 | Abcam | ab175658 | Immunostaining | 1:400 |
| anti mouse IgG-HRP | Amersham | NA931-1ml | Western | 1:10,000 |
| anti rabbit Alexa Fluor-488 | Invitrogen | A21206 | Immunostaining | 1:400 |
| anti rabbit Alexa Fluor-546 | Invitrogen | A10040 | Immunostaining | 1:400 |
| anti rabbit Alexa Fluor-594 | Invitrogen | A21207 | Immunostaining | 1:400 |
| anti rat Alexa Fluor-488 | Invitrogen | A21208 | Immunostaining | 1:400 |
| anti rat Alexa Fluor-594 | Invitrogen | A21209 | Immunostaining | 1:400 |
| anti rat Alexa Fluor-647 | Jackson ImmunoResearch | 712-605-153 | Immunostaining | 1:400 |
| CD19 | Abcam | ab245235 | Immunostaining | 1:100 |
| CD31 | R&D Systems | AF3628 | Immunostaining | 1:100 |
| CD31-APC | R&D | FAB3628A | FACS | 1:50 |
| CD3e | BioRad | MCA27690GA | Immunostaining | 1:100 |
| CD3e FITC | Thermo | 11-0031 | FACS | 1:50 |
| CD4 | Biolegend | BLD-100506 | Immunostaining | 1:100 |
| CD4-APC | Thermo | 17-0042-82 | FACS | 1:50 |
| CD45 | Becton Dickinson | 550539 | Immunostaining | 1:100 |
| CD45 APC-Cy7 | BD Pharmingen | 557659 | FACS | 1:50 |
| CD45 PE-Cyanine7 | Thermo | 25-0451-82 | FACS | 1:50 |
| CD45-PE | BD Pharmingen | 553081 | FACS | 1:50 |
| CD8a PE | BD Pharmingen | 553033 | FACS | 1:50 |
| F4/80 | AbD Serotec | MCA497 | Immunostaining | 1:100 |
| GFAP | Novus | NB100-53809 | Immunostaining | 1:100 |
| GFP | 2BScientific Ltd | GFP-1010 | Immunostaining | 1:100 |
| GLUT-1 | Millipore | 07-1401 | Immunostaining | 1:100 |
| Iba 1 | Novus Biologicals | NB100-1028 | Immunostaining | 1:100 |
| ICAM1 | Abcam | ab222736 | Immunostaining | 1:100 |
| IFNγ- PE-Cyanine7 | Biolegend | 505826 | FACS | 1:50 |
| IL-17A Alexa Fluor 488 | Biolegend | 506910 | FACS | 1:50 |
| Myelin basic protein | Abcam | ab7349 | Immunostaining | 1:100 |
| Olig 2 | R&D | AF2418 | Immunostaining | 1:100 |
| PDPN-PE | eBioscience | Dec 81 | FACS | 1:50 |
| Prox1 | ReliaTech | 102-PA32AG | Immunostaining | 1:100 |
| RFP | MBL | PM005 | Immunostaining | 1:100 |
| Tubulin | Sigma | T7451 | Western | 1:1000 |
| VE-Cadherin | R&D | AF1002 | Immunostaining | 1:100 |
| VE-Cadherin | Santa Cruz | sc-9989 | Immunostaining/western | 1:100/1:1000 |

cells were incubated overnight in a 5% CO2, incubator at 37 °C. Subsequently, cells were starved for 12 hours using starvation media. Slides were then connected to the flow system (Ibidi, 10902). Shear stress (15 dyn/cm²) was applied to cells for 12 hours, followed by an additional 6 hours of shear stress exposure with or without 1 µM A13.

### Transendothelial cell migration assay

$10^5$ HUVECs were seeded on Fibronectin-coated inserts of transwells (Corning, 3415) and cultured for 24-36 hours until they reached confluence. After the medium was then replaced by medium with or without 1 µM A13, cells were placed in the cell culture incubator for an

additional 48 hours. After removing the medium, $10^5$ human CD4 T cells in 0.5% BSA/RPMI medium were added to the insert, which was placed in a well filled with chemoattractant medium (50 ng/ml Cxcl12, R&D, 350-NS-010/CF and 0.5% BSA/RPMI). After six hours, cells in the suspension of the lower chamber were counted.

### Lentivirus generation and infection

The lentiviral APJ-GFP vector was purchased from Origene (RC207576L2) and coding sequences for tdTomato and APJ-T2A-GFP were inserted into the same lentiviral vector backbone. Vectors were expanded using DH5a cells and a Midiprep kit (Macherey-Nagel, MN740410.50). Lentiviruses were generated by transfecting 293 T cells with the lentiviral vector and packaging vectors (pMD2.G and psPAX2). The lentivirus-containing media were concentrated using Lenti X concentrator (Takara, PT4421-2) to 100x and stored at −80 °C. The concentrated lentiviruses were used to treat P1 HUVECs with 8 μg/mL polybrene overnight. The following day, the medium was replaced and the lentivirus-treated HUVECs were used for further assays and analysis.

### BALF cell isolation and blood cell contents measuring

Mice were fully anesthetized with Rompun and Ketamin before an incision was made in the neck to remove the salivary gland and expose the trachea. The head was fixed to straighten the trachea and intubation needles were inserted. Needles were slowly withdrawn while keeping the tube inside the trachea. The trachea and tube were tightly secured with nylon threads. A 1 ml syringe filled with 600 μl of DPBS was connected to the tube. DPBS was injected slowly and drained three times by manipulating the plunger, with the collected fluid being transferred to a 2 ml Eppendorf tube. This DPBS collection process was repeated three times. Blood was obtained from the abdominal vein and collected in a heparin-coated tube. The collected DPBS volume was measured using a pipette, and cell concentration was determined using a Luna cell counter. The BALF and blood cells were analyzed using the Scil Vet abc Plus+ system following the manufacturer's instructions.

### Knockdown of APLNR

For the downregulation of the apelin receptor in HUVECs, siAPLNR (Invitrogen, HSS100324) was used with siCont (Invitrogen, 12935200) serving as control. The transfection reagent was prepared by mixing 3.6 μL of siRNA (20pmol/μL) and 9 μL of RNAiMax (Invitrogen, 13778075) diluted in Opti-MEM before the mixture was incubated for 20 minutes. The transfection reagent was then added to $10^5$ adherent HUVECs and allowed to incubate overnight. The assay was conducted between 24-48 hours post-transfection.

### Bulk RNA seq and qPCR

RNA from HUVECs, tissues, and sorted cells was isolated using the RNeasy Plus Micro kit (Qiagen, 74034). For qPCR analysis, cDNA was synthesized using the iScript cDNA synthesis Kit (Bio-Rad, 1708891), and qPCR was carried out using TaqMan primers (listed below) and the SsoAdvanced Universal Probes Supermix (Bio-Rad, 1725284). For bulk RNA-seq, libraries were generated using NEBNext (NEB, E7760L), and sequencing was performed on an Illumina NextSeq 500 with mid output.

The following primers were used for qPCR: Cdh5 (Mm00486938_m1), Cldn5(Mm00727012_s1), Klf4(Mm00516104_m1), Apln(Mm00443562_m1), Aplnr(Mm00442191_s1), Icam1(Mm00516023_m1), Il6(Mm00446190_m1), Ccl2(Mm00441242_m1), Gja4(Mm01179783_m1), Actb(Mm00607939_s1).

### Bioinformatics analysis

The sequencing data in FASTQ format was processed using the BD Rhapsody WTA Analysis pipeline (version 1.0) on the SevenBridges Genomics online platform. The resulting expression matrix was used for further data analysis. Data normalization, dimensionality reduction, and visualization were performed using Seurat (version 4.3.0) unless specified otherwise.

For initial quality control of the extracted gene-cell matrices, cells were filtered using the following parameters: nFeature_RNA > 500 & nFeature_RNA < 6000 for the number of genes per cell, percent.mito <25 for the percentage of mitochondrial genes, and genes with parameter min.cells = 3. Filtered matrices were normalized using the LogNormalize method with a scale factor of 10,000. Variable genes were identified using the FindVariableFeatures function with the following parameters: selection.method = "vst", nfeatures = 2000, trimmed for the genes related to cell cycle (GO:0007049), and then used. Data integration was performed using the FindIntegrationAnchors and IntegrateData functions with default options. Statistically significant principal components were determined using the JackStraw method, and the first 9 principal components were used for UMAP non-linear dimensional reduction.

Unsupervised hierarchical clustering analysis was performed using the FindClusters function in the Seurat package. We tested different resolutions between 0.1 - 0.9 and selected the final resolution using the clustree R package to determine the most stable and relevant resolution for our previous knowledge. The cellular identity of each cluster was determined by finding cluster-specific marker genes using the FindAllMarkers function with a minimum fraction of cells expressing the gene over 25% (min.pct=0.25) and a log2 fold change threshold of 0.25 (logfc.threshold=0.25).

For subclustering analysis, specific cluster(s) were isolated using the subset function, and the data matrix was extracted from the Seurat object using the GetAssayData function. The whole analysis pipeline was then repeated from data normalization. The FeaturePlot, VlnPlot, and DoHeatmap functions of the Seurat package were used for visualization of selected genes. Gene set enrichment analysis (GSEA) was performed using the Molecular Signatures Database (MSigDB) v7.1 hallmark gene sets (mouse version) with the fgsea R package (version 1.24.0).

For bulk RNA-seq analysis, total RNAs were extracted using the RNeasy Plus Micro kit (Qiagen). The quality and quantity of RNA samples were analyzed with a Bioanalyzer and RNA 6000 pico kit (Agilent). Double-stranded cDNA was synthesized using the SMART-Seq v4 Ultra Low Input RNA kit for Sequencing (Takara), and sequencing libraries were constructed with the Nextera XT DNA Library Preparation Kit (Illumina). The resulting sequencing libraries were sequenced with 2 × 75 bp paired-end reads on the NextSeq 500 sequencer (Illumina). Sequenced reads were aligned to the human (hg38) reference genome using TopHat (version 2.1.1), and the aligned reads were used for transcript quantification by using HTSeq-count (version 0.6.1). DESeq2 (version 1.44.0) was used to identify differentially expressed genes across the samples.

### T cell isolation

In line with Dutch legislation and the Declaration of Helsinki, 50 mL peripheral whole blood was drawn from healthy volunteers. All volunteers provided informed consent and all protocols were approved by the Amsterdam University Medical Centre ethical committee (METC). Blood was processed within 2 hours of donation.

First, 10−20 mL of blood was mixed 1:1 with PBS + 5% TNC (trisodiumcitrate, Merck, 1.06447.5000) and gently layered on top of 12.5 mL Ficoll (Cytiva, 17144003) in a 50 mL tube. This tube was centrifuged at 800 x $g$ for 20 minutes with slow start and no brake. After centrifugation, the PBMC ring was pipetted out carefully and pipetted into a fresh 50 mL tube. The PMBC fraction was washed once in PBS + 5% TNC, after which they were centrifuged for 10 min at 300x g. Remaining erythrocytes were then lysed with 45 mL ice-cold lysis

buffer (155 mM NH4CL, 10 mM KHCO$_3$, 0.1 mM EDTA, pH7.4 in Milli-Q (Gibco, A1283-01) for 15 min at ice, with a 10 min 300 x $g$ centrifuge step after lysis. Cells were then resuspended in RT HEPES+ (20 mM HEPES, 132 mM NaCl, 6 mM KCL, 1 mM CaCl$_2$, 1 mM MgSO$_4$, 1.2 mM K2HPO4, 5 mM glucose (All Sigma-Aldrich), and 0.4% (w/v) human serum albumin (Sanquin Reagents), pH7.4) and total cell numbers were counted.

To isolate CD4+ T cells from the PBMC fraction, a negative selection separation kit (Miltenyi Biotec, 130-096-533) was used according to manufacturer's instructions. After isolation, CD4+ T cells were resuspended 2 mil/mL in RPMI 1640 medium (Gibco, 11875093) with 100 U/mL penicillin and streptomycin (P/S) and were kept in a 12 well plate overnight at 37 °C in 5% CO2.

### Flow assay
The protocol for the leukocyte flow assays is based on a previous study[90]. 30000 HUVECs, expressing either APJ-GFP or tdTomato, per lane were seeded into a FN-coated Ibidi μ-slides VI0.4 (Ibidi, Munich, Germany) and grown for 72 hours. Medium was refresh twice daily. 24 hours before the experiment, 1/1000 A13, or DMSO was added to the HUVECs. 4 hours before the experiment, 10 ng/mL LPS (Sigma, L2880) was added to all HUVEC to mimic inflammation. A13 and DMSO were refreshed also.

CD4 + T cells were resuspended 10$^6$/mL in 37°C HEPES+ and labelled using Vybrant™ DiD Cell-labeling solution (1/6000) for 20 minutes. The ibidi flow slides were connected to a perfusion system and underwent shear flow of 0.8 dyn/cm$^2$. Flow was turned on 3 minutes before 700.000 CD4+ T cells were injected upstream of the ibidi flow chamber. CD4 + T cell TEM (transendothelial migration) dynamics were recorded using an Axiovert 200 M widefield microscope, equipped with a 10x NA 0.30 DIC Air objective (Zeiss). Fluorescent excitation was induced by a HXP 120 X light source (100% intensity). For transmitting light, a TL Halogen Lamp at 6.06 V. An AxioCam Icc 3 (Zeiss) camera was used for detection. In the phase-contrast channel, an exposure of 32 ms was used. In the DiD/Far-red channel, an exposure of 3000 ms with an 625-655 excitation filter, a 660 beam splitter and a 655-715 emission filter were used.

To analyse CD4 + T cell crawling dynamics and preferred route of diapedesis, images were taken approximately every 5 seconds for 20 minutes at 3 positions in the middle of the ibidi lane. Immediately after time-lapse acquisition, tile-scans were performed to later quantify total adhesion and transmigration efficacy. Images were taken using Zen Blue software from Zeiss. The tile-scan was stitched in this software too, based on the phase-contrast image.

### Flow assay analysis
All analyses were performed in Imaris version (10.0.0) and were based on earlier publications[90]. Total adhesion and diapedesis efficacy were measured by performed a spot analysis on the tile-scans. Spots were detected based on DiD signal, with an estimated diameter of 8 μm. Spots were manually thresholded based on Imaris' quality parameter. Transmigrated and non-transmigrated CD4+ T cells were distinguished based on their intensity in the phase contrast channel, where non-transmigrated CD4+ T cells were white and transmigrated CD4+ T cells black. Total adhesion was calculated as # non-transmigrated CD4+ T cells + transmigrated CD4+ T cells. CD4+ T cell diapedesis efficacy was quantified as (# transmigrated CD4+ T cells) / (# non-transmigrated CD4+ T cells + transmigrated CD4+ T cells) * 100%.

To quantify crawling dynamics, the same spot analysis was performed on CD4+ T cells during the time-lapses. CD4+ T cells were detected up until their moment of diapedesis, again based on their intensity in the phase-contrast channel. An auto-regressive motion tracking algorithm in Imaris was added to connect all detected spots in the video, allowing a maximum distance of 20 μm between frames. A gap size (a frame without spot detected) of 1 was allowed in this

algorithm. All tracks with less than 4 spots were filtered out to ensure only proper crawling tracks were measured. Crawling speed, duration and length were extracted from these data.

In the time-lapses, we manually counted for each diapedesis event if it took place at a cell-cell junction (paracellular diapedesis), or through the cell body (transcellular diapedesis). Data were displayed as percentages for each donor.

### Serum sample collection and ELISA analysis from MS patients
A total of 108 serum samples were collected from 36 male and 72 female participants (Supplementary Data 1). Samples were divided into three groups based on health status: healthy donors, multiple sclerosis (MS) relapse, and MS remission. The study was approved by the local ethics committee Ärztekammer Westfalen-Lippe under the approval numbers 2016-053-f-S and 2010-262-f-S. All patients provided written consent. The human A13 level in the collected serum samples was examined using the CUSABIO ELISA Kit (CSB-E13072h) according to the manufacturer's instructions. Sample selection and serum analysis were done independently. Health status information was unblinded after ELISA assays.

### Statistics
Reproducibility was ensured by analyzing 3 independent samples or by conducting 3 independent experiments, unless indicated otherwise. No statistical method was used to predetermine sample size, and no animals were excluded from the analysis. All quantifications were performed using GraphPad Prism (GraphStats Technologies, version 9). For disease incidence and score, a two-way ANOVA with Geisser-Greenhouse correction was applied. Other quantifications were analyzed with a two-sided Student's $t$-test. Differential expression testing for DEG analysis was performed based on the two-sided Negative Binomial model for bulk RNA-seq (Figs. 2c and 6b) and on the two-sided non-parametric Wilcoxon rank sum test for scRNA-seq (Fig. 3f, Supplementary Fig. 10c, 11c, 12a). DEGs were visualized with volcano plots using the EnhancedVolcano R package (version 1.10.0).

## Data availability
The scRNA-seq data generated in this study have been deposited in the gene expression omnibus (GEO, https://www.ncbi.nlm.nih.gov/geo/) under the accession number GSE230551 (https://www.ncbi.nlm.nih.gov/geo/query/acc.cgi?acc=GSE230551). All other relevant data supporting the key findings of this study are available within the article and its Supplementary Information files. Source data are provided with this paper.

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

## Acknowledgements

We thank Martin Stehling (Max Planck Institute for Molecular Miomedicine, Germany) for cell sorting, our animal facility for excellent animal care, Eva Maria Schumann (Biobank management of Universitätsklinikum Münster, Germany) for serum preparation, Kristy Red-Horse (Stanford University) for Aplnr-CreERT2 mice, and Prof. Gou-Young Koh (Institute for Basic Science, Korea) for fruitful scientific discussions. This study was supported by the Max Planck Society, the University of Münster, the DFG (CRC 1366 and CRC 1009), and the Leducq Foundation.

## Author contributions

H.P., L.S., L.K., J.D.vB., and R.H.A. designed experiments and interpreted results. H.P., J.S., M.L.B.G., B.I.K., E.B., and K.P.K. conducted all

experiments including animal models, cell culture, imaging, quantifications, and analysis of human serum samples. H.P., B.I.K. and H.W.J. generated and analysed the scRNA-sequencing data. P.AK. generated critical genetic tools. H.P. and R.H.A. wrote the manuscript.

## Funding

## Competing interests
The authors declare no competing interests.
