## [Peer Review file · Nature Communications]

Apelin modulates inflammation and leukocyte recruitment in experimental autoimmune encephalomyelitis

Corresponding Author: Professor Ralf Adams

Version 0:

Reviewer comments:

Reviewer #1

(Remarks to the Author)

Summary:

The manuscript submitted by Park et al. attempts to elucidate a new functional aspect of the currently emerging lung-brain axis.

The authors demonstrate that Apelin treatment delays immune cell recruitment and entry into the lung in an LPS-induced model of pulmonary inflammation and during EAE, an animal model of multiple sclerosis in which the lung serves as a hub for autoreactive T cells on their way to the inflamed CNS.

Park et al. postulate that the underlying mechanism is an Apelin-induced internalization of its receptor in certain pulmonary endothelial cells, leading to downregulation of the expression of inflammatory chemokines and cell adhesion molecules critical for immune cell recruitment and their transmigration across the lung endothelium. They infer that in EAE, the impediment of this crucial step interferes with the immune cells' migratory route through the lung and recruitment to the inflamed CNS, leading to a delayed onset and ameliorated disease course.

Although the overall concept of the study is both novel and potentially interesting, the authors should clarify a number of open questions about conceptual aspects and conclusions drawn in order to build confidence in its validity, as the data in their current form do not support their hypothesis.

Major points:

- The authors demonstrate that the Apelin-induced internalization of its receptor results in the downregulation of the expression of inflammatory chemokines and adhesion molecules crucial for transendothelial migration (TEM). They conclude that therefore the immune cells are not able to enter the lung and further migrate to the inflamed CNS. While the data potentially suggest a certain delay, in principal CD45+ cells seem to be perfectly capable of entering the lung tissue and form clusters within the lung parenchyma. In fact, the highest number of immune cell clusters is observed in Apelin-treated animals at the peak of disease, which suggests that not their TEM-dependent entry but their egression from the lung might be impaired. Assuming the cells were entrapped in the vasculature, a higher number of them would be expected to be present in the blood of Apelin-treated animals and the formation of immune cell clusters would be expected to be generally reduced and not only delayed (and later even increased). Instead, the cells appear to be entrapped within the lung parenchyma, i.e., after having transmigrated through the endothelium. This important point is unclear and left largely undiscussed.

The main incongruences indicated above and further points are discussed in more detail below:

- The authors repeatedly claim that immune cell recruitment into the lung is delayed by A13. The only data that suggests such a delay, however, is Suppl. Fig. 1, where in the histological image provided the number of immune cell clusters in the lung on D7 appears to be lower in the A13-treated animals. The corresponding quantification is missing (Fig. 1f only displays EAE onset and peak, not D5 and D7). None of the other data suggests such a delay.
- The authors' main hypothesis is that immune cells fail to properly transmigrate through the lung endothelium because Apelin-treatment downregulates the expression of relevant adhesion molecules on lung ECs. However, the highest number of immune cell clusters is found in A13-treated animals at the peak of EAE, which shows that the cells were definitely able to enter the lung and indicates that they were trapped within the parenchyma (after TEM) rather than the vascular system (before TEM). Aplnr is also expressed on lymphatic endothelial cells and in theory A13 treatment could also interfere with the egress of the cells from the lung. It would be important to investigate and discuss this point.
- Aplnr is also expressed by lymphatic endothelial cells. The authors start the Apelin-treatment on the day of immunization, which means that it could potentially interfere with the priming phase and/or T cell egress from the lymph nodes. Thus, a

close examination of T cell numbers and functional state in the lymph nodes as well as of the endothelial cells of the lymph nodes would be important at early time points. Furthermore, in order to rule out an interference with the T cell priming phase and to confirm the specificity of the A13 effect on lung ECs, an EAE experiment should be performed in which the treatment is started after this crucial phase. Alternatively, a transfer EAE could be performed.

- The authors use the MOG35-55 peptide-induced “standard” model of EAE in C57Bl/6 mice. In this model, the spinal cord is the main target of the autoimmune attack while the brain is affected to a much lesser extent. All CNS-related data in the manuscript, however, are on the brain while ignoring the spinal cord.

- The authors claim that Apelin-treatment delays the onset and ameliorates the course of EAE. While the former is certainly true, the latter is less evident. The curve of the control group appears to have not reached the maximum score yet on day 16, the timepoint defined as “peak”. It would be interesting to see a full / longer EAE clinical course to see if the A13-treated group would eventually reach similar scores to the control group.

Minor points:

- The authors repeatedly (pages 3, 5, 11, 16, 17) claim that the activation of (autoreactive) T cells in the lung is an essential step in order for them to be fully pathogenic and able to enter the CNS. This is a misconception. It is true that T cells can be activated in the lung (intratracheal immunization with their cognate antigen). In the context of the paper, important for their ability to enter the CNS, however, is not their activation but their “reprogramming” from an activated to a migratory mode, which, in fact, includes the downregulation of activation markers.

- No sufficient information on the A13 treatment can be found in the “Materials and methods” section. In the “Mice and EAE mouse model” part, it only states: “Throughout the experiment, A13 or PBS was administered intraperitoneally to the mice, and body weight and disease severity were assessed daily.” In the legend of Figure 1, it is stated: “Female C57Bl/6 mice aged 8-12 weeks received intraperitoneal (i.p.) A13 or vehicle injections every other day commencing on the day of immunization.” Information like dose, manufacturer etc. are completely missing.

- In the flow assay, human CD4+ T cells are used. Phenotype and adhesion properties of those T cells are presumably very different from the ones of post-activatory CD4+ T cells entering the lung upon MOG-immunization. These differences should be considered when evaluating the result of the flow experiment.

- No information on the human samples is provided in the “Materials and methods” section. Some information is given in the corresponding figure legend (Supplementary Figure 18), while other information (e.g., age and disease characteristics) is completely missing.

- Several Mismatches between figures, legends and text, e.g.:
 - Legend Figure 1: letters in text do not correspond to letters in figure
 - Suppl. Figure 8: “c” is missing
 - Legend Suppl. Figure 15: “e” is missing (2x “d”)
 - Page 11, l. 262: Suppl. Fig. 2 is actually Suppl. Fig. 1

- Several spelling mistakes, e.g.:
 - Page 2, l. 28: remove “of”
 - Page 6, l. 124: “significantly”
 - Page 6, ll. 136/137: “lung” missing before microvascular ECs
 - Page 7, l. 168: “downregulated” instead of “downregulation”

- Furthermore, there is a number of statements whose logic is difficult to understand, e.g.:

Page 17: Our current study shows that Apelin-13 interferes with immune cell recruitment to the lung and, thereby, delays T cell entry into the CNS at an early phase of EAE before the development of disease symptoms. Later, a range of immune cell subpopulations accumulate and appear trapped in A13-treated lungs, which also results in reduction of their number in peripheral blood relative to vehicle-treated EAE animals. This course of events together with the low or absent expression of Apelin receptor in the adult CNS vasculature, [...] strongly argue for a role of the pulmonary vasculature in the A13-mediated protection against EAE.

- If the treatment interferes with immune cell recruitment to and entry into the lung, should not there be more immune cells in the blood and fewer cells in the lung?

Page 19: At the mechanistic level, we link the biological effect of A13 to reduced T cell adhesion to ECs (Fig. 5f, g). E-selectin, which is one of the critical TEM mediators in peripheral tissues that is downregulated by A13, functions in the rolling interactions that decelerate leukocytes and position them in close proximity to the endothelial monolayer. A13 also reduces the endothelial expression of ICAM-1 and VCAM-1, which mediate leukocyte adhesion through interactions with integrin receptors and are both upregulated by inflammation. Interestingly, previous work has shown that effector T cells are temporarily entrapped in the pulmonary vasculature on their way to lymph nodes during systemic inflammation. T cell entrapment in lung is reduced in mice lacking ICAM-1 and the related ICAM-2 molecule, whereas neutrophil recruitment is unaffected.

- The cells in this study seem to be trapped in the parenchyma rather than the vasculature. Also, if T cell entrapment is decreased in mice lacking ICAM-1 and ICAM-2, how do the authors explain that here the entrapment is highly increased in EAE +A13 mice, that is, a situation in which ICAM-1 is substantially downregulated on ECs?

Reviewer #2

(Remarks to the Author)

Apelin, the related peptide ligand Apela/Elabela (A13) and its receptor APJ/Aplnr regulate cardiovascular morphogenesis during development. Recent literature suggests this receptor-ligand pair also was involved in pathogenesis of cancer and inflammatory diseases. Here Authors tested the anti-inflammatory function of A13 peptide in the mouse EAE model for human MS. Authors reported treatment with A13 at time of immunization with MOG (day 0) reduced disease onset and severity (Fig 1). Unfortunately, a weakness is that experiments were ended at peak of disease instead of day 30. Nonetheless, Authors via a thorough set of experimental approaches clearly demonstrate the mechanism was through internalization and desensitization of its G-protein-coupled receptor APJ/Aplnr in the lung vascular endothelium thus limiting T cell release/extravasation from lung nesting sites to CNS. Well done!

Experiments employed A13 peptide administration and EC-specific inactivation of the Aplnr gene in adult (8 wk) mice in the EAE model and siRNA-mediated gene knockdown in in vitro models of cultured HUVECs for T cell adhesion and transendothelial migration. The protective effects of a genetic loss of Aplnr specifically in the endothelium in the EAE model was less robust than that of A13 peptide treatment. Interestingly, disease severity was much greater in the KO study (Fig 6e) compared to A13 Rx cohort (Fig 1a). Finally, Authors did not explore EAE protection in an Aplnr null WT BM chimeric animal to distinguish EC vs BM contributions, but these studies are beyond the scope of this study.

Comments/suggestions:

Major comment/critique: Although well done, mechanistic, and of interest, a crucial end point in this disease model was not used. Standard MOG model is immunization followed by a 28 - 30-day monitoring of disease. With your current data set Fig 1a ending at day-16, however, we don't know if protection by A13 was simply a delay.

Other comments/suggestion:

1. Results Fig 1a: Comment from above summary. What is the EAE score in A13 treated vs control mice days 16 -30? Authors cannot assume that day16 is peak in A13 cohort. Could be that lung educated T cells eventually exit, traffic to CNS/spinal cords and cause pathology and disease, or worse, remain in lung and cause lung pathology! In addition, I encourage Authors to treat a cohort of mice at peak EAE disease with A13 peptide to test if A13 has a therapeutic impact. This c/would address an important idea/possibility regarding therapeutic value of blocking this pathway in MS a chronic, relapsing-remitting disease.
2. Authors cite literature that found the lung acts as a nesting ground for T cells during initiation of disease in EAE. As you appreciate, there also is significant literature that spleen and draining LN are key to generate pathogenic T cells in EAE. Yet, these organs were not examined, or the data was not included. If you have these data, please consider adding to Fig 2. Could this observation in lung be explained by differences in the time point analyzed? That is, the dLNs initially, and then spleen, followed by passage through lung on their way to the CNS? These concepts could add to the Discussion.
3. Introduction Lines 75-78: The nomenclature as presented is confusing. Is Apelin-13 the same aa sequence as Apelin, which Authors called the related peptide ligand Apela/Elabela?
4. The authors should identify what region of brain (cerebellum is commonly used in EAE) and what was EAE disease scores (range is best) of mice in all Figs that included brain tissue images.

Minor

1. Fig 1b-c, 1e-f legends (there is no Fig 1g) are not correct; Line 1015. Insert "d" to identify the heatmap data. "or A13-treated HUVEC. [d]"; Pg 42, Line 1080. Should include the flow amount applied; Line 1085, need to add in legend that white color cells = CD45 signal; Lines 503-504 state "After 18 hours, animals were sacrificed and lungs were prepared for further processing." Authors present data taken at 4 and 24 hrs not 18 hrs. Methods, legend and Figs should agree. Line 152: What time point was image of lungs- 4 or 24hr? "Immunofluorescence staining revealed that A13 reduces the number of CD45+ cells in lungs of treated compared to non-treated mice (Supplementary Fig. 5a). Line 262 "immunization (Supplementary Fig. 2)." Authors probably mean supplemental fig 1. Line 264 Authors probably mean Fig 1f, there is no Fig 1g.
2. Misspelled line 65, PSGL1 (P-Selektin Glykoprotein Ligand-1); line 89 administration (administration); line 358 treatment (Supplementary Fig. 6), siAPLNR reduces the lw baseline expression.
3. Fig 5f-h does not provide the actual % of adherent T cells that have migrated across the HUVEC monolayers. Mention HUVEC are treated with LPS in text. Should be included here as panel Fig 5i is static exp, not flow study.

Reviewer #3

(Remarks to the Author)

In this manuscript, Park et al. showed delayed EAE onset and reduced disease severity in mice treated with the peptide ligand Apelin 13 (A13).

The authors showed that A13 interferes with immune cell recruitment to the lung reducing T cell activation in the periphery and thereby entry into the CNS. In the same line, delayed EAE seems to be associated with a delayed accumulation of immune cells within the lung.

The effects of Apelin-13 appear to be mediated by the expression modulation of cell adhesion molecules by the peptide. This would lead to a modulation of inflammatory responses in lung endothelial cells and an altered transendothelial migration of immune cells into the lung. The delayed effects were explained by the Apelin-mediated internalization and desensitization of its G protein-coupled receptor APJ/Aplnr.

Although the effect and mechanism of action of A13 treatment to endothelial cells has been clearly characterized and defined in this paper, the results does not fully support the statement that A13 effects on EAE “can be attributed to the disruption of a critical role of the lung in neuroinflammation.”

Indeed, based on the data presented here, a direct A13 effect on the CNS vasculature or other peripheral organs such as lymph nodes or the intestine cannot be discarded.

Main comments:

1. The introduction contemplates only publications on EAE pathogenesis that cannot be extrapolated to MS. If the authors aim to comment on MS in the Introduction, please refer to other current reviews on MS pathophysiology, which is indeed much more complex than EAE pathogenesis.

2. The pre-clinical efficacy of A13 administration has been demonstrated in a unique EAE experiment including 5 mice/group. The treatment effect appears to be certainly very pronounced. However, validation experiments including a proper numbers of animals is needed. Differences between mice that under treatment did not develop EAE and those with a delayed onset will be interesting but impossible with n=5.

In addition, a treatment control group (naïve + A13) is needed to assess A13 effects in the absence of neuroinflammation

3. The authors indicate that Apelin receptors were not expressed in healthy brain tissue. However, to completely exclude that AP13 treatment of EAE mice could have a direct brain effect, receptor expression on inflamed brain tissue during EAE should be evaluated.

Indeed, although no differences were observed in lung immune cell numbers at onset, the CNS already showed decrease numbers of immune cells infiltration suggesting alternative mechanisms of action of A13.

4. During A13 treatments, no differences in infiltrated immune cells in the lung were observed at onset; however, lower numbers of immune cells were detected in the peripheral blood of A13-treated mice. Does the delayed infiltration into the lung cause the sudden decrease in peripheral blood cells? or, are other organs such as the intestine (where apelin receptor is also expressed) partially responsible of this lower immune cell numbers in peripheral blood?

In this line, a potential implication of the intestine in this experimental setup should be considered.

5. In the EC-specific Aplnr knockouts (Aplnr^{ECKO}), the authors assume that ICAM-1 expression is not affected in the EAE. However, it is well established that brain endothelial cells increase ICAM-1 expression during EAE. Thus, ICAM-1 expression in the brain needs to be evaluated too. In this line, a direct effect of the knockout on brain endothelial cells should be considered and studied.

6. How is the lung weight at D5 compared to that of D0 or unimmunized mice? Could it be possible that A13 induces lung inflammation, accumulation of fluids and therefore increases the weight? I cannot be excluded that A13 may cause lung inflammation/ fibrosis and affect respiratory function.

In this line, the authors indicate in Lines 254-257 – “fibrosis, analysis of immunostained tissue sections reveals only a slight increase in Collagen type I and no accumulation of alpha-smooth muscle actin (SMA) in the immune cell cluster-containing areas of EAE and A13-treated samples relative to naïve lung”

However, only a representative IF image is not enough to state that. A quantitative analysis is needed to support this conclusion.

General comments:

- No information is provided on imaging and imaging analysis protocols although several figures (Fig 1c, Fig 2a, Fig 3b, ...) show quantification of images data. How many images/region or which area were quantified? In Fig 1c for example, the percentage of CD45+/CD31 cells is in the whole brain or within the image plotted in Fig 1b? Which brain regions were investigated? Where different regions quantified?

- Figures are not always numbered in the order they appear in the text.

- Some of the supplementary figures would also benefit from quantitative analysis to support conclusions (in particular Sup Fig 1, 4b, 12).

Minor comments:

- [Line 89] – misspelling “administration”

- [Line 101] – EAE symptoms were detected in all vehicle treated mice at day 13, not day 12 based on the graph.

- [Line 124]- misspelling “significantly”.

- [Lines 154-156] – “relative to controls, while immune cell numbers in peripheral blood are not significantly changed (Supplementary Fig. 5b and c).”

However, an effect cannot be properly assessed here, considering the variability in the data points and the small sample

size (n=5).

- [Line 255]- misspelling “increase”.
- [Figure Legend 1] – It does not correspond to Figure 1 or the text. Adjust to both. Figure 1c shows for instance quantification of CD45, and not ICAM as indicated in Results.
- Fig. 4e is not commented/mentioned
- [Figure 1b] – Images are not very informative on the overall cell infiltration into brain, I suggest providing images at a lower magnification, where brain immune infiltration along CNS regions can be observed and compared.
- [Supplementary figure legend 2]- The title “Effect of A13 on EAE brain and circulating immune cells” does not correspond to the figure, in which lung weights and circulating immune cells comparison are depicted. How many animals per group were included in this data set?
- [Lines 323-327] the authors mentioned that effects of A13 on ICAM expression were investigated. However, no data on ICAM is depicted in the corresponding Fig 5.
- [Lines 409-411] – “Later, a range of immune cell subpopulations accumulate and appear trapped in A13-treated lungs, which also results in reduction of their number in peripheral blood relative to vehicle-treated EAE animals.” This sentence does not completely reflect the data, lower peripheral blood immune cell numbers were observed at onset, while no differences in immune cell numbers were observed in the lungs.

Author Rebuttal letter:

First of all, we would like to thank all reviewers for their constructive comments and criticisms, which have enabled us to improve the manuscript substantially. As you will see, changes involve the inclusion of new results but also numerous improvements in data presentation and interpretation. In our view, it is now much more straightforward to understand how we have reached certain conclusions and we have also conducted a number of experiments to exclude potential alternative mechanisms.

While a detailed point-by-point response to each individual comment is provided further below, we would like to start with a summary of the most important new results:

New experiments with 8 animals per group confirm that the effect of A13 on EAE incidence and disease symptoms persists up to day (D) 30 (Fig. 1a). The resulting data also confirm that D16 corresponds to the peak of EAE both in the vehicle control and A13 groups. Overall, considering the data in the original submission and the revised manuscript, our findings are now based on a robust number of more than 20 animals per cohort.

We have added additional time points in our analysis of immune cell accumulation inside the A13-treated lung (Supplementary Fig. 2a, b). These results now clearly show that A13 delays immune cell entry in early EAE (at D7) relative to vehicle control mice. Apelin-13 also delays the resolution of immune cell clusters later in the course of the EAE experiment so that A13-treated lungs contain more CD45+ clusters at D16. However, this difference is no longer visible at D30.

Several comments indicated that the reviewers were concerned about potential effects of A13 on the early immune response to MOG35-55 immunization. Newly added data shows that that two different therapeutic treatment regimes in which A13 is given from D7 or D11 onward can significantly reduce disease score (Fig. 7a), which also raises the possibility that the ligand might be useful for therapeutic intervention in inflammatory diseases. Furthermore, our scRNA-seq analysis of immune cells from lung shows similar increases in activation markers in T cells from EAE +vehicle and EAE +A13 animals compared to naïve animals (Supplementary Fig. 12a-c).

Other new data show that A13 does not alter immune cell numbers in inguinal lymph nodes (near the site of MOG35-55 injection) at D7 (Supplementary Fig. 1c), which further argues against an effect on early T cell responses. In addition, new FACS data show that A13 treatment reduces immune cell numbers in mediastinal lymph nodes (near the lung) at D11 (Supplementary Fig. 1a), which is consistent with the data showing reduced levels of immune cells in peripheral blood at D11 (Supplementary Fig. 3b). In contrast, we see no difference in vehicle control and A13-treated spleens at D11 or D16 (Supplementary Fig. 1a, b). Taken together, these results support the conclusion that A13 acts by delaying or impairing immune cell entry into the lung but also trafficking through the lung into lymph nodes and peripheral blood.

1

Several reviewers had raised questions about the reasons for the delayed immune cell trafficking that is overt in A13-treated lungs at EAE peak. There are actually very few published studies addressing T cell migration inside inflamed lungs, but it has been suggested that blood vessels play a role in this process (Mrass et al., Nat. Commun. 2017 PMID:

29044117). While we are not able to fully elucidate how T cell trafficking inside the lung parenchyma is influenced by A13, we have included new data indicating reduced T cell polarization (Fig. 4f, g). In this context, we would also like to emphasize that the A13-induced alterations in endothelial cells are not limited to ICAM-1 but also affect other cell adhesion molecules, certain cytokines and multiple additional regulators of inflammation (Fig. 2b-e), which can affect aspects of T cell migration such as cell polarity and motility.

Other new data show that A13 does not induce immune cell accumulation in the lung of naïve animals (Supplementary Fig. 2c).

Another important set of new results confirms that our previous findings showing that A13 reduces neuroinflammation in the brain cortex also apply to spinal cord (Supplementary Fig. 1a, b) and cerebellum (Fig. 7b, d).

As requested, we have conducted some analysis of intestine and did not see overt alterations in response to A13 (Supplementary Fig. 6a, b).

Reviewers' comments:

Reviewer #1 (Remarks to the Author):

Summary:

The manuscript submitted by Park et al. attempts to elucidate a new functional aspect of the currently emerging lung-brain axis.

The authors demonstrate that Apelin treatment delays immune cell recruitment and entry into the lung in an LPS-induced model of pulmonary inflammation and during EAE, an animal model of multiple sclerosis in which the lung serves as a hub for autoreactive T cells on their way to the inflamed CNS.

Park et al. postulate that the underlying mechanism is an Apelin-induced internalization of its receptor in certain pulmonary endothelial cells, leading to downregulation of the expression of inflammatory chemokines and cell adhesion molecules critical for immune cell recruitment and their transmigration across the lung endothelium. They infer that in EAE, the impediment of this crucial step interferes with the immune cells' migratory route through the lung and recruitment to the inflamed CNS, leading to a delayed onset and ameliorated disease course. Although the overall concept of the study is both novel and potentially interesting, the authors should clarify a number of open questions about conceptual aspects and conclusions drawn in order to build confidence in its validity, as the data in their current form do not

2

support their hypothesis.

Major points:

• The authors demonstrate that the Apelin-induced internalization of its receptor results in the downregulation of the expression of inflammatory chemokines and adhesion molecules crucial for transendothelial migration (TEM). They conclude that therefore the immune cells are not able to enter the lung and further migrate to the inflamed CNS.

While the data potentially suggest a certain delay, in principal CD45+ cells seem to be perfectly capable of entering the lung tissue and form clusters within the lung parenchyma. In fact, the highest number of immune cell clusters is observed in Apelin-treated animals at the peak of disease, which suggests that not their TEM-dependent entry but their egression from the lung might be impaired. Assuming the cells were entrapped in the vasculature, a higher number of them would be expected to be present in the blood of Apelin-treated animals and the formation of immune cell clusters would be expected to be generally reduced and not only delayed (and later even increased). Instead, the cells appear to be entrapped within the lung parenchyma, i.e., after having transmigrated through the endothelium. This important point is unclear and left largely undiscussed.

Thank you very much for your comment. We have clarified this important point in the revised manuscript. At least two distinct phases need to be considered in the lung after the initiation of EAE: 1) immune cell infiltration and accumulation in clusters in early EAE; and 2) immune cell egress and cluster resolution, which is visible in controls at peak EAE (see Supplementary Fig 2a, b). Our results in the revised manuscript show that Apelin-13 delays both steps so that immune cell clusters are only visible at D11 (in contrast to D7 in control) and clusters persist at peak EAE when a profound reduction has already occurred in vehicle control animals. As a consequence, the number of immune cells in peripheral blood and in mediastinal lymph nodes is reduced. This, in turn, strongly reduces the entry of pathogenic cells into the CNS, resulting in delayed EAE progression and milder disease symptoms

relative to vehicle-treated EAE animals.

Regarding the underlying mechanism, A13 reduces the endothelial expression of ICAM-1 and VCAM-1 – two mediators that facilitate leukocyte adhesion through interactions with integrin receptors. A13 also lowers expression of the proinflammatory cytokines CCL2 and IL6, which are known to contribute to immune cell transmigration through the endothelium. Unfortunately, very little is known about the cellular and molecular mechanisms immune cell trafficking through the lung. One of the very few publications addressing this question has shown that intravenously injected GFP+ effector T cells switch between two migration modes and migrate along the vasculature in the lung of LPS-treated mice (PMID: 29044117). While full clarification of the processes controlling T cell migration in the lung and the impact of A13 would require extensive further research that is beyond the scope of this paper, we have added new data showing that T cell polarity might be altered in A13 lungs (Fig. 4f, g

3

of the revised manuscript). Furthermore, our scRNA-seq data uncover changes that affect the expression of chemokines. Accordingly, the persistence of immune cell clusters in the lung of A13-treated mice at peak EAE might be caused by the altered cytokine expression, the delayed entry of immune cells into the lung or impaired migration towards lymphatic vessels. Regarding the latter, it is noted that various studies have revealed important roles of Apelin signaling in the growth and remodeling of the lymphatic vasculature (Tatin et al., JCI Insight 2017, PMID: 28614788; Kim et al., Arterioscler Thromb Vasc Biol. 2014, PMID: 24311379; Berta et al., Sci Rep. 2021, PMID: 33707612).

The main incongruences indicated above and further points are discussed in more detail below:

â€ The authors repeatedly claim that immune cell recruitment into the lung is delayed by A13. The only data that suggests such a delay, however, is Suppl. Fig. 1, where in the histological image provided the number of immune cell clusters in the lung on D7 appears to be lower in the A13-treated animals. The corresponding quantification is missing (Fig. 1f only displays EAE onset and peak, not D5 and D7). None of the other data suggests such a delay.

Thank you for this comment. As suggested, we have now added data for the early time points and quantification to Supplementary Fig. 2. The data in the revised manuscript now clearly show that A13 treatment reduces the number and size of CD45+ cell clusters at D7, whereas significantly more cluster are retained in the lung at D16 (EAE peak). The intermediate data point, namely D11, already shows a trend in the same direction but the differences are not yet statistically significant.

â€ The authorsâ main hypothesis is that immune cells fail to properly transmigrate through the lung endothelium because Apelin-treatment downregulates the expression of relevant adhesion molecules on lung ECs. However, the highest number of immune cell clusters is found in A13-treated animals at the peak of EAE, which shows that the cells were definitely able to enter the lung and indicates that they were trapped within the parenchyma (after TEM) rather than the vascular system (before TEM). Aplnr is also expressed on lymphatic endothelial cells and in theory A13 treatment could also interfere with the egress of the cells from the lung. It would be important to investigate and discuss this point.

Thank you for this comment. As already mentioned above, A13 not only affects the expression of cell adhesion molecules but also chemokines and other factors that are known to be relevant for leukocyte migration and inflammation. We trust that the inclusion of additional results and improvements in data presentation clarify that A13 delays immune cell recruitment to the lung but also affects later trafficking processes and thereby the resolution of immune cell clusters later in EAE. Whether the latter involves compromised interactions with lymphatic vessels is unclear, but it should be noted that our scRNA-seq data indicates

4

very low Aplnr expression in pulmonary lymphatic ECs (LECs) and very limited A13-induced changes in LEC gene expression. (Supplementary Fig. 9b). Moreover, Aplnr-CreERT2-mediated Cre reporter activation fails to label lung lymphatics. We also see minimal variations in the gene expression profiles of T cells from EAE +vehicle and +A13 conditions (Supplementary Fig 12a-c), which argues against direct effects of A13 on these lymphocytes.

Taken together, we agree that T cell trafficking and egress from the lung are compromised and we have revised the text to reflect this important aspect of the phenotype.

â€¢ Apelin is also expressed by lymphatic endothelial cells. The authors start the Apelin-treatment on the day of immunization, which means that it could potentially interfere with the priming phase and/or T cell egress from the lymph nodes. Thus, a close examination of T cell numbers and functional state in the lymph nodes as well as of the endothelial cells of the lymph nodes would be important at early time points. Furthermore, in order to rule out an interference with the T cell priming phase and to confirm the specificity of the A13 effect on lung ECs, an EAE experiment should be performed in which the treatment is started after this crucial phase. Alternatively, a transfer EAE could be performed.

Thank you very much for this comment. Lymphatic endothelial cells are already covered in our answer to the previous question.

Following the reviewer's suggestions, we have conducted a number of additional experiments. Treatment with Apelin-13 during the course of EAE development â€” namely administration on every second day starting at D7 or daily injection starting at D11 â€” both successfully reduced disease score (Fig. 7a). We also provide further data for the second condition (A13 daily starting at D11) showing the accumulation of CD45+ cells in the lung as well as reduced presence of CD45+ in the brain and lower ICAM-1 expression in the cerebellum at D20 (Fig. 7b-d). These new findings are fully consistent with our previous results and also rule out that the protective effect of A13 in EAE is caused by a compromised immune response in the early phase after MOG35-55 administration. Gene expression differences between T cells from EAE +vehicle and +A13 conditions are minimal (Supplementary Fig 12a-c), which also argues against the possibility that A13 compromises T cell activation.

â€¢ The authors use the MOG35-55 peptide-induced â€”standardâ€” model of EAE in C57Bl/6 mice. In this model, the spinal cord is the main target of the autoimmune attack while the brain is affected to a much lesser extent. All CNS-related data in the manuscript, however, are on the brain while ignoring the spinal cord.

Thank you for this comment. We now show FACS data for immune cell populations in spinal cord at D11 and D16 (Supplementary Fig. 1a, b). Importantly, the D16 results for spinal cord

5
are fully consistent with our findings in the brain and show that A13 profoundly reduces all immune cell populations analyzed. At D11 (EAE onset), we see a significant reduction (around 50%) of CD45 cells, whereas T cell subpopulations were not yet detectable in spinal cord samples from both in the EAE +vehicle and EAE +A13 groups.

â€¢ The authors claim that Apelin-treatment delays the onset and ameliorates the course of EAE. While the former is certainly true, the latter is less evident. The curve of the control group appears to have not reached the maximum score yet on day 16, the timepoint defined as â€”peakâ€”. It would be interesting to see a full / longer EAE clinical course to see if the A13-treated group would eventually reach similar scores to the control group.

Agree. We now show the full course of EAE progression up to D30 in Fig. 1a of the revised manuscript. Importantly, this confirms that the control group consistently reaches peak disease symptoms at D16-D17. Moreover, the peaks for disease incidence and score also occur around D16-D17 in the A13 group and the beneficial effects persist until D30 (Fig. 1a).

Minor points:

â€¢ The authors repeatedly (pages 3, 5, 11, 16, 17) claim that the activation of (autoreactive) T cells in the lung is an essential step in order for them to be fully pathogenic and able to enter the CNS. This is a misconception. It is true that T cells can be activated in the lung (intratracheal immunization with their cognate antigen). In the context of the paper, important for their ability to enter the CNS, however, is not their activation but their â€”reprogrammingâ€” from an activated to a migratory mode, which, in fact, includes the downregulation of activation markers.

Thank you very much for this feedback. We fully agree with this comment and have modified the text accordingly.

â€¢ No sufficient information on the A13 treatment can be found in the "Materials and methods" section. In the â€”Mice and EAE mouse modelâ€” part, it only states: â€”Throughout the

experiment, A13 or PBS was administered intraperitoneally to the mice, and body weight and disease severity were assessed daily. In the legend of Figure 1, it is stated: "Female C57Bl/6 mice aged 8-12 weeks received intraperitoneal (i.p.) A13 or vehicle injections every other day commencing on the day of immunization." Information like dose, manufacturer etc. are completely missing.

Thank you very much for alerting us to this issue. We have updated the Methods section and have, among other changes, added the missing information on A13:

6

Throughout the experiment, 100 μ l of A13 (100 μ M, Bio-technie, #2420) or PBS was administered intraperitoneally to the mice, and body weight and disease severity were assessed daily

In the flow assay, human CD4⁺ T cells are used. Phenotype and adhesion properties of those T cells are presumably very different from the ones of post-activatory CD4⁺ T cells entering the lung upon MOG-immunization. These differences should be considered when evaluating the result of the flow experiment.

We agree that there are obvious limitations associated with ex vivo/in vitro experiments, which are relevant for this type of approach in general. Human CD4⁺ T cells from peripheral blood are frequently utilized in adhesion and transendothelial cell migration assays, often in conjunction with human umbilical vein endothelial cells (HUVECs). Furthermore, the expression of chemokine receptors and other molecules is different between activated and naive CD4⁺ T cells. In addition, CD4⁺ T cells constitute a heterogeneous population, including subsets such as Th1, Th2, Th17, and $\gamma\delta$ T cells, each exhibiting different levels of integrin expression.

Despite of these limitations, important information can be gained from in vitro experiments exactly because of their simplicity and accessibility. For example, naive CD4⁺ T cells and activated Th cells but also many other leukocyte populations express Integrin alpha L (Itgal), the binding partner of ICAM-1 (see scRNA-seq data below). Thus, A13-induced downregulation of ICAM-1 in endothelial cells is likely to be relevant for many of these cell types.

[image redacted]

No information on the human samples is provided in the "Materials and methods" section. Some information is given in the corresponding figure legend (Supplementary Figure 18), while other information (e.g., age and disease characteristics) is completely missing.

Thank you very much for alerting us to this issue. We now provide the relevant patient metadata in Excel format. This includes information about age, sex, diagnosis, therapy and types of diseases.

7

Several Mismatches between figures, legends and text, e.g.:

Legend Figure 1: letters in text do not correspond to letters in figure > corrected

Suppl. Figure 8: "â" is missing > corrected

Legend Suppl. Figure 15: "â" is missing (2x "â") > corrected

Page 11, l. 262: Suppl. Fig. 2 is actually Suppl. Fig. 1 > corrected

Several spelling mistakes, e.g:

Page 2, l. 28: remove "of" > corrected

Page 6, l. 124: "significantly" > corrected

Page 6, ll. 136/137: "lung" missing before microvascular ECs > corrected

Page 7, l. 168: "downregulated" instead of "downregulation" > corrected

Furthermore, there is a number of statements whose logic is difficult to understand, e.g.:

Page 17: Our current study shows that Apelin-13 interferes with immune cell recruitment to the lung and, thereby, delays T cell entry into the CNS at an early phase of EAE before the development of disease symptoms. Later, a range of immune cell subpopulations accumulate and appear trapped in A13-treated lungs, which also results in reduction of their number in peripheral blood relative to vehicle-treated EAE animals. This course of events together with the low or absent expression of Apelin receptor in the adult CNS vasculature, [â] strongly argue for a role of the pulmonary vasculature in the A13-mediated protection against EAE.

â If the treatment interferes with immune cell recruitment to and entry into the lung, should not there be more immune cells in the blood and fewer cells in the lung?

Thank you for this feedback. In general, we have improved data presentation, interpretation and discussion in the revised manuscript, which should address many of the points that were not clear in the original submission. Regarding the question at the end of this comment, there is actually a slight increase in the count of CD45+ immune cells from peripheral blood of A13-treated EAE mice at D7, but this difference is very small compared to the difference that is seen in the opposite direction at D11 (EAE onset) (Supplementary Fig. 3b).

Page 19: At the mechanistic level, we link the biological effect of A13 to reduced T cell adhesion to ECs (Fig. 5f, g). E-selectin, which is one of the critical TEM mediators in peripheral tissues that is downregulated by A13, functions in the rolling interactions that decelerate leukocytes and position them in close proximity to the endothelial monolayer. A13 also reduces the endothelial expression of ICAM-1 and VCAM-1, which mediate leukocyte adhesion through interactions with integrin receptors and are both upregulated by inflammation. Interestingly, previous work has shown that effector T cells are temporarily entrapped in the pulmonary vasculature on their way to lymph nodes during systemic

8
inflammation. T cell entrapment in lung is reduced in mice lacking ICAM-1 and the related ICAM-2 molecule, whereas neutrophil recruitment is unaffected.

â The cells in this study seem to be trapped in the parenchyma rather than the vasculature. Also, if T cell entrapment is decreased in mice lacking ICAM-1 and ICAM-2, how do the authors explain that here the entrapment is highly increased in EAE +A13 mice, that is, a situation in which ICAM-1 is substantially downregulated on ECs?

Thank you very much for your comment. This point has been already addressed by the previous answers. In essence, we show that A13 delays immune cell entry into the lung and accumulation in clusters in early EAE. Subsequent steps of immune cell trafficking from the lung into the circulation and cluster resolution are also delayed, which is most obvious at peak EAE (see Supplementary Fig 2a, b).

Reviewer #2 (Remarks to the Author):

Apelin, the related peptide ligand Apela/Elabela (A13) and its receptor APJ/Aplnr regulate cardiovascular morphogenesis during development. Recent literature suggests this receptor-ligand pair also was involved in pathogenesis of cancer and inflammatory diseases. Here Authors tested the anti-inflammatory function of A13 peptide in the mouse EAE model for human MS. Authors reported treatment with A13 at time of immunization with MOG (day 0) reduced disease onset and severity (Fig 1). Unfortunately, a weakness is that experiments were ended at peak of disease instead of day 30. None-the-less, Authors via a thorough set of experimental approaches clearly demonstrate the mechanism was through internalization and desensitization of its G-protein-coupled receptor APJ/Aplnr in the lung vascular endothelium thus limiting T cell release/extravasation from lung nesting sites to CNS. Well done!

Experiments employed A13 peptide administration and EC-specific inactivation of the Aplnr gene in adult (8 wk) mice in the EAE model and siRNA-mediated gene knockdown in in vitro models of cultured HUVECs for T cell adhesion and transendothelial migration. The protective effects of a genetic loss of Aplnr specifically in the endothelium in the EAE model was less robust than that of A13 peptide treatment. Interestingly, disease severity was much greater in the KO study (Fig 6e) compared to A13 Rx cohort (Fig 1a). Finally, Authors did not explore EAE protection in an Aplnr null WT BM chimeric animal to distinguish EC vs BM contributions, but these studies are beyond the scope of this study.

Comments/suggestions:

Major comment/critique: Although well done, mechanistic, and of interest, a crucial end point in this disease model was not used. Standard MOG model is immunization followed by

9

a 28 - 30-day monitoring of disease. With your current data set Fig 1a ending at day-16, however, we don't know if protection by A13 was simply a delay.

Thank you very much for the helpful feedback, which is most appreciated. As suggested, we have extended the EAE experiment and followed disease development until day 30 (D30). The resulting new data confirm that the peak of EAE symptoms occurs around D16 for both vehicle control and A13 animals (Fig. 1a). Furthermore, disease incidence and score were persistently reduced by A13 until D30. We have also added new data showing that A13 is also beneficial when the treatment is started at D7 or D11 (Fig. 7a, b). These results rule out that the beneficial effects of A13 are caused by the disruption of the initial immune response after MOG administration. Furthermore, our findings show that A13 has therapeutic potential in the EAE model with possible relevance for neurological autoimmune diseases.

Regarding EAE protection in an *Aplnr* null WT BM chimeric animal to distinguish EC vs BM contributions, we would like to point that the manuscript include results with EC-specific *Aplnr* mutant mice (Fig. 6d-g). We appreciate that a potential contribution of Apelin receptor in hematopoietic cells is not ruled out by this approach. However, according to the *Tabula muris* scRNA-seq database, *Aplnr* expression is confined to ECs. Likewise, expression data for hematopoietic cells (haemosphere.org) shows only low expression of *Aplnr* (much lower than, for example, *Ccxc4*) and no expression in the B cell or T cell lineages (see below).

[image redacted]

Other comments/suggestion:

1. Results Fig 1a: Comment from above summary. What is the EAE score in A13 treated vs control mice days 16 -30? Authors cannot assume that day16 is peak in A13 cohort. Could be

10

that lung educated T cells eventually exit, traffic to CNS/spinal cords and cause pathology and disease, or worse, remain in lung and cause lung pathology! In addition, I encourage Authors to treat a cohort of mice at peak EAE disease with A13 peptide to test if A13 has a therapeutic impact. This c/would address an important idea/possibility regarding therapeutic value of blocking this pathway in MS a chronic, relapsing-remitting disease.

Thank you very much for your comments. As suggested, we have extended our analysis of EAE development to D30 and thereby confirmed that the peak of EAE occurs around D16 and that the positive effect of A13 persist until the end of the experiment. The revised manuscript contains new data for spinal cord (Supplementary Fig. 1a, b), which shows a strong A13-induced reduction of immune cell entry that is comparable to the effect seen in brain. Regarding potential negative consequences of the persistence of immune cell clusters in the lung of A13-treated mice at peak EAE, we see no increase in collagen type I and α -SMA immunostaining relative to vehicle-treated EAE mice, even though there is a notable increase in collagen I in both groups relative to naïve mice (Supplementary Fig. 14a, b).

As already mentioned further above, we have also added new data showing that A13 is also beneficial when the treatment is started is D7 or D11 (Fig. 7a, b). These results rule out that the effect of A13 is caused by the disruption of the initial immune response after MOG administration and they also indicate that future work should explore the therapeutic potential of Apelin-13. We have not investigated the impact of starting the treatment at peak EAE because there is a profound decrease of disease score and incidence even in vehicle-treated (control) animals between D16 and D30 (Fig. 1a). This is an inherent limitation of the transient EAE model, which does not mimic all aspects of chronic and recurring neuroinflammatory disease in humans.

2. Authors cite literature that found the lung acts as a nesting ground for T cells during initiation of disease in EAE. As you appreciate, there also is significant literature that spleen

and draining LN are key to generate pathogenic T cells in EAE. Yet, these organs were not examined, or the data was not included. If you have these data, please consider adding to Fig 2. Could this observation in lung be explained by differences in the time point analyzed? That is, the dLNs initially, and then spleen, followed by passage through lung on their way to the CNS? These concepts could add to the Discussion.

Thank you very much for this helpful suggestion and, indeed, we are, of course, aware that lymphoid organs play a central role in the activation of pathogenic T cells prior to their reprogramming into a migratory mode and further trafficking.

We are also aware that there are publications arguing that T cell licensing can occur in a variety of organs and that adoptively transferred Th cells are much more abundant in the spleen than in the lung (Tan et al., J Immunol. 2017, PMID: 27986906). In our own experiments, however, FACS analyses show no significant differences in CD45+ cells or

11

various T cell subsets in the spleen of EAE mice treated with vehicle or A13 (Supplementary Fig. 1a, b). We also see no overt differences in intestine (Supplementary Fig. 6a, b), which was previously shown to influence EAE (reviewed by Parodi and Kerlero de Rosbo, Front Immunol. 2021, PMID: 34621267).

As suggested, we have the revised text of the manuscript to address the complexity of the processes during EAE.

3. Introduction Lines 75-78: The nomenclature as presented is confusing. Is Apelin-13 the same aa sequence as Apelin, which Authors called the related peptide ligand Apela/Elabela?

We are sorry about this confusion. Apelin is proteolytically processed, converting a larger 77-amino acid (aa) (in humans) preproprotein into various shorter products including a biologically active 13-aa isoform. This peptide is further activated by a pyroglutamyl modification at its N-terminus, which generates the more potent [Pyr1]-Apelin-13. [Pyr1]-Apelin-13 is also what we have used in this manuscript (termed A13 or Apelin-13 for brevity).

Apela/Elabela is encoded by a separate gene and is a second ligand for the APJ/Aplnr receptor.

We made sure that the information above is provided in the revised manuscript.

4. The authors should identify what region of brain (cerebellum is commonly used in EAE) and what was EAE disease scores (range is best) of mice in all Figs that included brain tissue images.

Thank you very much for this comment. In our initial study, we have analyzed immune cell infiltration into the cortex in the data shown in Fig. 1b. In addition, we have included new results showing the cerebellum of mice treated with A13 at D11 and analysis at D20 (Fig 7d). Importantly, results from cortex and cerebellum are fully consistent. Information for disease score has been added to the legends for Figures 1b (EAE is 3, +A13 is 1) and 7d (EAE is 2, +A13 is 1).

Minor

1. Fig 1b-c, 1e-f legends (there is no Fig 1g) are not correct; > corrected
Line 1015. Insert – to identify the heatmap data. – A13-treated HUVEC. [d];
Pg 42, Line 1080. Should include the flow amount applied; > corrected
Line 1085, need to add in legend that white color cells = CD45 signal; > corrected
Lines 503-504 state –After 18 hours, animals were sacrificed and lungs were prepared for

12

further processing.– Authors present data taken at 4 and 24 hrs not 18 hrs. Methods, legend and Figs should agree. ; > corrected

Line 152: What time point was image of lungs- 4 or 24hr? –Immunofluorescence staining revealed that A13 reduces the number of CD45+ cells in lungs of treated compared to non-treated mice (Supplementary Fig. 5a). > corrected

Line 262 immunization (Supplementary Fig. 2). Authors probably mean supplemental fig 1.). > corrected

Line 264 Authors probably mean Fig 1f, there is no Fig 1g. > corrected

2. Misspelled line 65, PSGL1 (P-Selektin Glykoprotein Ligand-1); > corrected

line 89 administration (administration); > corrected

line 358 treatment (Supplementary Fig. 6), siAPLNr reduces the lw baseline expression.

> corrected

3. Fig 5f-h does not provide the actual % of adherent T cells that have migrated across the HUVEC monolayers. Mention HUVEC are treated with LPS in text. Should be included here as panel Fig 5i is static exp, not flow study.

Thank you very much for these suggestions. In our experiments, around 30% of adherent T cells migrated through HUVEC monolayers (see graph below). This ratio is not substantially altered by the overexpression of Aplnr or A13 treatment, which supports that the changes in T cell transmigration mainly reflect altered adhesion to ECs.

The transendothelial cell migration assay (Fig. 5i) was performed under static condition without activation of HUVEC by LPS, the T cell adhesion assay (Fig. 5f-h) was performed under flow with LPS challenge to induce ICAM-1 expression. We made sure that LPS pretreatment is indicated in the corresponding figure legend.

13

Reviewer #3 (Remarks to the Author):

In this manuscript, Park et al. showed delayed EAE onset and reduced disease severity in mice treated with the peptide ligand Apelin 13 (A13).

The authors showed that A13 interferes with immune cell recruitment to the lung reducing T cell activation in the periphery and thereby entry into the CNS. In the same line, delayed EAE seems to be associated with a delayed accumulation of immune cells within the lung. The effects of Apelin-13 appear to be mediated by the expression modulation of cell adhesion molecules by the peptide. This would lead to a modulation of inflammatory responses in lung endothelial cells and an altered transendothelial migration of immune cells into the lung. The delayed effects were explained by the Apelin-mediated internalization and desensitization of its G protein-coupled receptor APJ/Aplnr.

Although the effect and mechanism of action of A13 treatment to endothelial cells has been clearly characterized and defined in this paper, the results does not fully support the statement that A13 effects on EAE can be attributed to the disruption of a critical role of the lung in neuroinflammation.

Indeed, based on the data presented here, a direct A13 effect on the CNS vasculature or other peripheral organs such as lymph nodes or the intestine cannot be discarded.

Thank you very much for your feedback and for highlighting open questions. As suggested by the reviewer, we have explored potential alternative explanations for the effect of A13. First of all, we confirmed that A13 does not interfere with T cell priming in the early phase of EAE development. The number of CD45+ cells and various T cell subpopulations is not altered in the inguinal lymph node near the site of MOG35-55 injection (Supplementary Fig. 1c). The same populations are also not altered in spleen at D11 (EAE onset) or D16 (EAE peak) (Supplementary Fig. 1a, b). Moreover, treatment with Apelin-13 during the course of EAE development namely administration on every second day starting at D7 or daily injection starting at D11 both successfully reduced disease score (Fig. 7a). We also provide further data for the second condition (A13 daily starting at D11) showing the accumulation of CD45+ cells in the lung as well as reduced presence of CD45+ in the brain and lower ICAM-1 expression in the brain at D20 (Fig. 7b-d). These new findings are fully consistent with our previous results and indicate that A13 might allow therapeutic intervention in an early phase of disease development with potential relevance for MS.

We have also analyzed CD45+ cells and various T cell subpopulations in the mediastinal lymph node near the lung (Supplementary Fig. 1a, b). Here we saw a significant reduction in immune cells at D11 (EAE onset), which is consistent the reduction of the same populations in peripheral blood (Supplementary Fig. 3b), which was already included in the original

submission. In contrast, we found no overt differences in intestine (Supplementary Fig. 6a,

14

b), which was previously shown to influence EAE (reviewed by Parodi and Kerlero de Rosbo, Front Immunol. 2021, PMID: 34621267).

We appreciate that it is very difficult to completely rule out effects through organs other than the lung. Nevertheless, the sum of our all our results â including gene expression data and the analysis of additional tissues â strongly argues that A13 reduces EAE incidence and disease score through its action on the pulmonary endothelium, which impairs immune cell trafficking through the lung.

Main comments:

1. The introduction contemplates only publications on EAE pathogenesis that cannot be extrapolated to MS. If the authors aim to comment on MS in the Introduction, please refer to other current reviews on MS pathophysiology, which is indeed much more complex than EAE pathogenesis.

We agree that MS is much more complicated than EAE and have made sure that this is reflected by the introduction as well as the rest of the manuscript. Current reviews on MS pathophysiology have been included as references. In addition, we emphasize differences between EAE and MS both in the Introduction and the Discussion.

2. The pre-clinical efficacy of A13 administration has been demonstrated in a unique EAE experiment including 5 mice/ group. The treatment effect appears to be certainly very pronounced. However, validation experiments including a proper numbers of animals is needed. Differences between mice that under treatment did not develop EAE and those with a delayed onset will be interesting but impossible with n=5. In addition, a treatment control group (na⁻ve + A13) is needed to assess A13 effects in the absence of neuroinflammation

Thank you very much for these comments. The effect of A13 is robustly supported by multiple data sets with more than 20 mice per group in total. Fig. 1a of the revised manuscript shows new results with 8 mice per group and analysis over 30 days after immunization (Fig. 1a). Fig. 1a of the original submission showed a separate set of results up to D16 with 6 mice per group. Further experiments, conducted for FACS analysis and immunostainings at intermediate time points, follow the same trend as the longer time course (see graphs on the left below). Thus, our findings are based on a robust number of experiments involving several independent cohorts of animals and different batches of A13.

As requested, we have also tested the effect of A13 (treatment for 16 days) on the lung of na⁻ve animals. We could not see any change in the number of CD45+ immune cells (see

15

panels on the right below). If one also considers the results of the LPS challenge (Supplementary Fig. 7a-c), it is clear that A13 reduces lung inflammation rather than inducing it.

4 100 100
EAE EAE EAE
80
+A13 80 +A13 +A13
3
Incidence (%)
Disease score

Incidence (%)
60 60
2
40 40

1 20 20

0 0 0

D2 D4 D6 D8 D10 D12 D14 D16 D0 D2 D4 D6 D8 D10 D12 D14 D16 D0 D2 D4 D6 D8 D10 D12
D0 D2 D4 D6 D8 D10 D12
Days after immunization Days after immunization Days after immunization
Days after immunization

4 100 100

EAE EAE EAE

80

+A13 80 +A13 +A13

3

Incidence (%)

Disease score

Incidence (%)

60 60

2

40 40

1 20 20

0 0 0

D2 D4 D6 D8 D10 D12 D14 D16 D0 D2 D0

D4 D2

D6 D4

D8 D6

D10 D8

D12D10

D14D12

D16D14 D16 D0 D2 D4 D6 D8 D10 D12 D14 D16

Days after immunization Days after immunization Days after immunization

3. The authors indicate that Apelin receptors were not expressed in healthy brain tissue. However, to completely exclude that AP13 treatment of EAE mice could have a direct brain effect, receptor expression on inflamed brain tissue during EAE should be evaluated. Indeed, although no differences were observed in lung immune cell numbers at onset, the CNS already showed decrease numbers of immune cells infiltration suggesting alternative mechanisms of action of A13.

We agree that there is the possibility that *Aplnr* expression is increased in brain ECs during EAE. However, one of our previous studies has analyzed the effect of EAE on different cell populations in the brain by scRNA-seq (Jeong et al., eLife 2022, PMID: 36197007). Examination of the endothelial cells in the published data confirmed limited expression of *Aplnr* in a few *Cdh5+* ECs of the healthy cortex and no increase (perhaps even downregulation) in EAE (see below and Supplementary Fig. 5e).

In addition, we have clarified the time course of immune cell entry into vehicle control and A13-treated lungs (Supplementary Fig. 2a, b). The results show that CD45+ cell clusters are strongly reduced in A13 lungs at D7 after MOG immunization, indicating that Apelin-13 affects immune cell entry into the lung several days before EAE symptoms emerge.

16

4. During A13 treatments, no differences in infiltrated immune cells in the lung were observed at onset; however, lower numbers of immune cells were detected in the peripheral blood of A13-treated mice. Does the delayed infiltration into the lung cause the sudden

decrease in peripheral blood cells? or, are other organs such as the intestine (where apelin receptor is also expressed) partially responsible of this lower immune cell numbers in peripheral blood?

In this line, a potential implication of the intestine in this experimental setup should be considered.

Thank you very much for your feedback. We now provide a more detailed time course of the changes in the lung and show that the reduction of CD45+ immune cell clusters is already visible at day 7 after MOG immunization (Supplementary Fig. 2a, b). This precedes the reduction of several immune cell population in peripheral blood and in mediastinal lymph node near the lung at D11 (Supplementary Fig. 1a, b). As requested, we have also included intestine in our analysis and found no overt changes in CD45+ cells or tissue organization at D11 (EAE onset) and D16 (EAE peak).

Taken together, all our data argue that A13 delays immune cell trafficking through the lung and thereby suppresses the entry of encephalitogenic T cells into the CNS, leading to reduced EAE incidence and disease score.

5. In the EC-specific *Aplnr* knockouts (*Aplnr*ECKO), the authors assume that ICAM-1 expression is not affected in the EAE. However, it is well established that brain endothelial cells increase ICAM-1 expression during EAE. Thus, ICAM-1 expression in the brain needs to be evaluated too. In this line, a direct effect of the knockout on brain endothelial cells should be considered and studied.

Thank you for this comment. Levels of ICAM-1 in the brain of EAE mice are reduced by A13 treatment (Fig. 1b). Consistent with previous publications, we have also seen and reported that endothelial ICAM-1 is upregulated during EAE (Jeong et al., *eLife* 2022, PMID: 36197007). Furthermore, scRNA-seq data in Fig. 3d of the revised manuscript shows that EAE induces a strong upregulation of *Icam1* transcripts relative to naïve animals, whereas A13 treatment downregulates endothelial *Icam1*.

We have not investigated ICAM-1 expression in *Aplnr*ECKO brain because there is little reason to assume that the cell adhesion molecule would be altered in a relevant way in a setting of absent or very low neuroinflammation. Arguing further against a potential direct effect of A13 on the brain endothelium, we have added new data showing that *Aplnr* expression is very low/absent both in brain ECs from healthy adults but also EAE animals

17

(Supplementary Fig. 5e). Nevertheless, we show downregulation of ICAM-1 in the mutant pulmonary endothelium (Fig. 6c, d), which is consistent with our findings in A13-treated animals and in cell culture.

6. How is the lung weight at D5 compared to that of D0 or unimmunized mice? Could it be possible that A13 induces lung inflammation, accumulation of fluids and therefore increases the weight? I cannot be excluded that A13 may cause lung inflammation/ fibrosis and affect respiratory function.

In this line, the authors indicate in Lines 254-257 that fibrosis, analysis of immunostained tissue sections reveals only a slight increase in Collagen type I and no accumulation of alpha-smooth muscle actin (SMA) in the immune cell cluster-containing areas of EAE and A13-treated samples relative to naïve lung.

However, only a representative IF image is not enough to state that. A quantitative analysis is needed to support this conclusion.

Lung weight in EAE +vehicle control or EAE +A13 mice at D5 does not show any significant differences to naïve controls (see graph below). This is fully consistent with the observation that immune cell clusters in EAE +vehicle control lungs mostly emerge after D5, which is further delayed by A13 (Supplementary Fig. 2a, b). While we see that A13 interferes with immune cell trafficking through the lung, leading to a transient accumulation of CD45+ cells at D16 (EAE peak), it is noteworthy that these clusters resolve eventually and there is no longer a difference to vehicle control at D30 (Supplementary Fig. 2a, b).

We note that EAE induces a moderate upregulation of collagen type I at the sites of immune cell clusters (quantitation is now included in Supplementary Fig. 14a), there are no signs of widespread SMA upregulation, which is a feature of pulmonary fibrosis (e.g. Kulkarni et al., *Proteomics*. 2016, PMID: 26425798; Mahmoudi et al., *Adv. Pharm. Bull.* 2020, PMID:

32002364; Suliman et al., iScience 2021, PMID: 34977500). Consistent with our staining results, A13-treated animals did not show signs of respiratory distress.

[image redacted]

General comments:

18

â€¢ No information is provided on imaging and imaging analysis protocols although several figures (Fig 1c, Fig 2a, Fig 3b, â€¦) show quantification of images data. How many images/region or which area were quantified? In Fig 1c for example, the percentage of CD45+/CD31 cells is in the whole brain or within the image plotted in Fig 1b? Which brain regions were investigated? Where different regions quantified?

We apologize for omitting important details regarding methodology and quantifications. The relevant information has been added to relevant sections and figure legends.

Our initial analysis of immune cell infiltration into the brain was done on cortex, but we have added new results for cerebellum at D20 (Fig. 7b, d) and spinal cord at D11 and D16 (Supplementary Fig. 1a, b). All these results very consistently show that A13 administration protects the CNS against immune cell entry in EAE.

â€¢ Figures are not always numbered in the order they appear in the text. â€º corrected

â€¢ Some of the supplementary figures would also benefit from quantitative analysis to support conclusions (in particular Sup Fig 1, 4b, 12). â€º we have added quantitation whenever this was feasible

Minor comments:

â€¢ [Line 89] â€¢ misspelling â€¢administrationâ€¢ â€º corrected

â€¢ [Line 101] â€¢ EAE symptoms were detected in all vehicle treated mice at day 13, not day 12 based on the graph. â€º corrected

â€¢ [Line 124]- misspelling â€¢significantlyâ€¢. â€º corrected

â€¢ [Lines 154-156] â€¢ â€¢relative to controls, while immune cell numbers in peripheral blood are not significantly changed (Supplementary Fig. 5b and c).â€¢

However, an effect cannot be properly assessed here, considering the variability in the data points and the small sample size (n=5).

There is a statistically significant difference at D11 and newly added FACS data shows reduced immune cell numbers in mediastinal lymph nodes (near the lung) at D11. Taken together, our results that the entry of immune cells into the lung and subsequent trafficking processes are impaired by A13.

â€¢ [Line 255]- misspelling â€¢increaseâ€¢. â€º revised to â€¢marginal elevationâ€¢

â€¢ [Figure Legend 1] â€¢ It does not correspond to Figure 1 or the text. Adjust to both.

Figure 1c shows for instance quantification of CD45, and not ICAM as indicated in Results.

19

â€º corrected

â€¢ Fig. 4e is not commented/mentioned â€º this figure is mentioned in line 305 of the revised manuscript

â€¢ [Figure 1b] â€¢ Images are not very informative on the overall cell infiltration into brain, I suggest providing images at a lower magnification, where brain immune infiltration along CNS regions can be observed and compared.

Thank you for your comment. It is very difficult or impossible to show immune cells at low magnification. The revised manuscript contains FACS data for spinal cord (Supplementary Fig. 1b, c), which confirm observations in the brain cortex. Furthermore, we have added new

data for cerebellum from a separate set of A13 treatment experiments (Fig. 7b, d). The sum of these results is fully consistent with the EAE incidence and disease score data.

â€ [Supplementary figure legend 2]- The title â€Effect of A13 on EAE brain and circulating immune cellsâ€ does not correspond to the figure, in which lung weights and circulating immune cells comparison are depicted. How many animals per group were included in this data set?

Thank you for this feedback. We agree and have changed the title of the legend (now Supplementary Figure 3) to â€Effect of A13 on EAE lung weight and circulating immune cells.â€ The number of mice (5 for each condition) is now indicated in the legend.

â€ [Lines 323-327] the authors mentioned that effects of A13 on ICAM expression were investigated. However, no data on ICAM is depicted in the corresponding Fig 5.

Thank you for this comment. Reduced ICAM-1 expression in A13-treated animals is shown by immunostaining in Fig. 1b (brain) as well as Fig. 2a and Fig. 6b (lung). This finding is further supported by qPCR data and results in cell culture (Fig. 2b, c, e; Supplementary Fig. 8 and Supplementary Fig. 18a).

â€ [Lines 409-411] â€Later, a range of immune cell subpopulations accumulate and appear trapped in A13-treated lungs, which also results in reduction of their number in peripheral blood relative to vehicle-treated EAE animals.â€ This sentence does not completely reflect the data, lower peripheral blood immune cell numbers were observed at onset, while no differences in immune cell numbers were observed in the lungs.

We trust that we have clarified this important point in the revised manuscript. Reduction of CD45+ immune cell clusters in the lung is visible at day 7 after MOG immunization (Supplementary Fig. 2a, b), whereas the reduction of several immune cell populations in

20

peripheral blood is only visible at D11. Moreover, comparable numbers of the same cell populations were found in inguinal lymph nodes near the site of MOG injection at D7, which also indicates that A13 is not interfering with the early response to immunization. This last point is further supported by new results showing the beneficial effect of A13 treatment starting at later time points during EAE development (Fig. 7a, b).

21

Version 1:

Reviewer comments:

Reviewer #1

(Remarks to the Author)

In the revised version of their manuscript "Apelin is a critical modulator of inflammation and leukocyte recruitment", Park et al. have succeeded in significantly improving both the quality and the soundness of their study. Important points of criticism raised after the last submission have been convincingly addressed by rewriting relevant text passages and by conducting critical experiments, the results of which are in line with and further support their hypothesis. Congratulations on a nice study and an interesting new concept!

Reviewer #2

(Remarks to the Author)

The revised manuscript containing significant new data has addressed each of my comments. I have no further comments.

Reviewer #4

(Remarks to the Author)

Apelin – Ralf Adams

In this revised manuscript the authors provide convincing evidence for a role of Apelin/Apelin receptor interaction at the vascular level in the pathogenesis of EAE. The study shows that Apelin-13 ameliorates MOG35-55 induced EAE in C57BL/6 mice and changes the presence of immune cells in the lung. It is furthermore shown that Apelin induces internalization and thus desensitization of its GPCR receptor Aplnr in endothelial cells and reduces inflammatory gene expression in endothelial cells. In vitro data show an involvement of Aplnr in mediating T cell arrest to endothelial cells under flow. Importantly, endothelial cell specific deletion of Apelin receptor function also ameliorated EAE. The authors propose that their study shows a role for vascular Apelin/Aplnr interaction in mediating the migration of encephalitogenic T cells through the lungs as prerequisite for their subsequent trafficking to the CNS where they induce EAE.

Unfortunately, also the revised study as it stands does not allow to draw this conclusion. The authors base their study on the assumption that encephalitogenic T cells require reprogramming in the lung prior to entering the CNS and cause EAE. However, the studies that have described this sequence of events are based on transfer EAE studies in the Lewis rat where in vitro activated myelin-specific T cell blasts are intravenously injected into syngeneic recipients and due to their blast stage – which is prior to the migratory stage – are trapped in lung capillaries which is followed by their extravasation and maturation to migratory cell in the lung and mediastinal lymph nodes. The present study however investigates active EAE where MOG35-55-specific T cells are activated in the draining lymph nodes of the CFA /MOG deposit and from there migrate into the entire body of these mice including the lung, but also gut or meninges as shown by numerous publications. As induction of active EAE by the present approach also induces T cells specific for mycobacterial proteins and employs PTX and furthermore the present study unfortunately lacks a CFA control group the cellular and molecular underpinnings leading to CD45+ immune cell accumulation in the lung in this EAE model are very difficult to explain.

The study thus does not provide any direct evidence for the trafficking of encephalitogenic T cells via the lung in vivo and thus an effect of A13 treatment on ameliorating EAE due to delaying the trafficking of encephalitogenic T cells via the lung to the CNS.

The revised manuscript now includes additional timepoints investigating immune cell accumulation in the lung after EAE induction, however the authors still do not provide any data allowing to determine if these differences are due to immune cell entry, proliferation or trapping.

The revised study also does not sufficiently include investigation of encephalitogenic T cell trafficking and further activation in other mucosal sites such as the lamina propria of the small intestine and colon which are reported sites of encephalitogenic T cell activation (PMID: 37467267, doi.org/10.1016/j.celrep.2019.09.002) rather than the intestinal villi. Due to the assumption of the central role of the lung in EAE pathogenesis the authors unfortunately limited themselves and their data interpretation solely on this organ and did not consider that A13 might also delay T cell entry into the CNS via the vasculature of the choroid plexus lacking a BBB or via the leptomeningeal vessels. In fact Supplementary Figure 5 shows a prominent GFP signal in the Aplnr-GFP-reporter mouse in the meninges which might reflect high Aplnr expression on meningeal vessels. The authors do not describe if there is a GFP reporter signal for Aplnr in the choroid plexus vasculature or in the meninges of the spinal cord.

These missing answers could be obtained by injecting a low number of fluorescently tagged naïve 2D2 cells prior to EAE induction and after CFA injection and following their migration in vivo into different organs. Alternatively, flow cytometry analysis of the immune cells accumulating in the lung employing MHC class II MOG35-55 tetramers would allow to determine the presence of encephalitogenic T cells in the lung versus other mucosal and meningeal sites. In vivo imaging of the lung and the brain/spinal cord would also allow to obtain further insights.

Technical concerns affecting data interpretation

Several immunostainings of the lung and the brain are very confusing. Figure 1d shows CD45 staining of lung sections – this staining should show a vast amount of alveolar macrophages already in a healthy lung, which are not visible. Similarly, Figure 1b depicts ICAM-1 staining on vascular structures but not on CD45+ immune cells, which also express high levels of ICAM-1 and thus should be brightly stained. Furthermore activated microglial cells stain positive for CD45 in brains of mice suffering from EAE – again this staining is not visible in this figure. These examples raise concerns about the threshold settings during imaging or image analysis.

The gating strategy shown in Supplementary Figure 21 lacks a figure legend and raises many questions on how the different immune cell subsets isolated from the different organs were analyzed and quantified throughout the manuscript. The gating strategy needs to be revised/corrected prior to reanalysis of the flow cytometry data. Specifically, a figure legend explaining the gating strategies is lacking and needs to be included. At present the gate referred to as lymphocyte gate includes 3 cellular subsets- what are these subsets? The CD45 gate to the right includes CD45 low cells, which is CNS resident microglial cells rather than CNS infiltrating immune cells – it seems the authors did not consider this population appropriately. The gate depicted as “CD3 cells” seems to be almost random as it depicts two populations with high CD3 cell surface expression, one population with very low CD3 cell surface expression, one negative population and one population with intermediate CD3 immunofluorescence that is cut in half by the shown gating strategy. It is also not clear if the gating shows a sequential gating. In panel C the “lymphocyte gate” is not a gate but rather a live cell gate, the CD4 gate depicts small and large cells with the larger CD4+ cells.

Additional points:

Introduction:

The introduction still contains a number of inaccuracies with respect to MS, EAE and immune cell trafficking.

The etiology of MS is not known thus it is not appropriate to write “MS is mediated by autoreactive Th1 and Th17 cells but also involves B cells”. In fact there is also evidence for a contribution of CD8 T cells in MS pathogenesis and environmental factors contributing to MS are not even mentioned. Furthermore it remains to be shown if autoaggressive T cells in MS see myelin antigens and EAE can also be induced with encephalitogenic T cells that do not recognize myelin antigens (see PMID: 30787438).

As outlined above the authors base their work on the assumption that “It is also established that the generation of encephalitogenic T cells capable of penetrating the blood-brain barrier involves activation steps in lymph nodes, spleen and lung” and “research using EAE as an animal model has revealed an unexpected role of T cell reprogramming in the lung, which induces a migratory phenotype and enables the trafficking of these cells into the CNS”. While in active EAE the priming, activation and first proliferation of encephalitogenic T cells in secondary lymphoid organs is a given – the involvement of T cell activation steps in the lung is not an established concept in the sense that this is prerequisite for EAE pathogenesis. Rather, several studies have provided evidence for a gut-brain axis. The involvement of T cell “reprogramming” steps in the lung and mediastinal lymph nodes as prerequisite for their migration into the CNS has been observed in a transfer EAE model in Lewis rats where in vitro activated myelin specific T cell blasts are iv injected into syngeneic recipients. The model used in the present study is however very different as here encephalitogenic T cells are primed in vivo in the draining lymph nodes of the deposit of the CFA.

The authors are advised to be more accurate when referring to T cells or other immune cell subsets in the context of immune cell trafficking. It is not correct that at the BBB $\alpha 4$ -integrin interactions with VCAM-1 replace the function of PSGL-1 and its selectin ligands. Selectins are upregulated at the inflamed BBB and mediate immune cell rolling (see in vivo studies of the Constantin, Kubes, Engelhardt laboratories). Integrins have been shown to mediate slow rolling but rather are involved in the subsequent steps of immune cell arrest, polarization and diapedesis. Also GPCR signaling induced by chemokine binding activates all integrins expressed by immune cells rather than specifically $\alpha 4$ -integrins.

Results- just several examples for further improvement given:

Several statements about EAE are incorrect in the manuscript. The authors eg. mention that C57BL/6 mice suffering from EAE are resistant to further induction of EAE – as MOG35-55 induced EAE is a chronic disease one cannot induce EAE again in these mice as they will remain with mild symptoms until sacrificed. I guess the authors rather referred to the well known “adjuvans effect” : it was indeed shown that EAE cannot be induced by active immunization in mice after CFA immunization. These observations highlights again why the CFA control in the present manuscript would be of utmost importance for data interpretation.

A potential effect of A13 in relapsing-remitting or spontaneous EAE could be readily investigated in different EAE models available.

The authors show that in vitro A13 inhibits T cell adhesion to the endothelium however they repeatedly mention that A13 interferes with leukocyte transendothelial migration – but there is no data showing that A13 regulates immune cell diapedesis across HUVECs or the vascular wall. Please note immune cell extravasation is a multi-step process in which each step is mediated by distinct molecules – thus the data should be reported accurately. Studying the effect of A13 on 2D2 T cell migration across mouse primary lung endothelial cells compared to brain microvascular endothelial cells would have been a very elegant way to show involvement of A13 in regulating encephalitogenic T cell trafficking across the lung versus brain endothelium.

Figure 1 a: The authors refer to novel experiments with 8 mice per group in their reply to the reviewers – but this figure lacks error bars and mention of the number of animals investigated, the number of experiments performed and included into this figure. Thus it remains unclear if the study is underpowered and in addition reproducibility and variability of the observations described is not documented. These data need to be included and statistical analysis of a series of EAE experiments is best performed by comparing AUC for the clinical disease of all animals included using the statistical method of SuperPlots (Lord, S. J., Velle, K. B., Dyché Mullins, R. and Fritz-Laylin, L. K., Superplots: communicating reproducibility and variability in cell biology. *J. Cell Biol.* 2020. 219: e202001064.). All EAE experiments shown in this study should include provide this information.

Figure 1e – based on the Supplementary Figure 21 the gating strategies chosen for identifying immune cell subsets in the CNS have not considered CD45^{low} tissue resident microglial cells versus CD45^{high} infiltrating immune cells. These flow cytometry studies need to be reanalyzed and corrected.

Figure 2 states that A13 treatment downregulated ICAM-1 on endothelial cells but not epithelial cells – however the figure does not show any landmark allowing to determine ICAM-1 staining on epithelial cells.

Figure 5f – para and transcellular diapedesis cannot be distinguished without a landmark

Methods:

EAE scoring method used is not explained

Minor

The authors mix referring to leukocytes and immune cells – I propose to stick with the word “immune cell” as a summary as many researchers – and especially in the cell migration field use “leukocyte” to refer granulocytes.

Orientation of tissue sections should be explained, e.g. orientation of the brain sections shown – are those coronal, transversal or sagittal sections.

Author Rebuttal letter:

REVIEWER COMMENTS

Reviewer #1 (Remarks to the Author):

In the revised version of their manuscript “Apelin is a critical modulator of inflammation and leukocyte recruitment”, Park et al. have succeeded in significantly improving both the quality and the soundness of their study. Important points of criticism raised after the last submission have been convincingly addressed by rewriting relevant text passages and by conducting critical experiments, the results of which are in line with and further support their hypothesis. Congratulations on a nice study and an interesting new concept!

We thank the reviewer for the kind comments and the helpful feedback, which has allowed us to improve the manuscript substantially.

Reviewer #2 (Remarks to the Author):

The revised manuscript containing significant new data has addressed each of my comments. I have no further comments.

We are grateful for this assessment and the helpful input during the review process.

Reviewer #3:

This reviewer was no longer available to review the manuscript and was therefore replaced by Reviewer 4.

Reviewer #4 (Remarks to the Author):

In this revised manuscript the authors provide convincing evidence for a role of Apelin/Apelin receptor interaction at the vascular level in the pathogenesis of EAE. The study shows that Apelin-13 ameliorates MOG35-55 induced EAE in C57BL/6 mice and changes the presence of immune cells in the lung. It is furthermore shown that Apelin induces internalization and thus desensitization of its GPCR receptor Aplnr in endothelial

1

cells and reduces inflammatory gene expression in endothelial cells. In vitro data show an involvement of Aplnr in mediating T cell arrest to endothelial cells under flow. Importantly, endothelial cell specific deletion of Apelin receptor function also ameliorated EAE. The authors propose that their study shows a role for vascular Apelin/Aplnr interaction in mediating the migration of encephalitogenic T cells through the lungs as prerequisite for their subsequent trafficking to the CNS where they induce EAE.

Unfortunately, also the revised study as it stands does not allow to draw this conclusion. The authors base their study on the assumption that encephalitogenic T cells require reprogramming in the lung prior to entering the CNS and cause EAE. However, the studies that have described this sequence of events are based on transfer EAE studies in the Lewis rat where in vitro activated myelin-specific T cell blasts are intravenously injected into syngeneic recipients and due to their blast stage – which is prior to the migratory stage – are trapped in lung capillaries which is followed by their extravasation and maturation to migratory cell in the lung and mediastinal lymph nodes. The present study however investigates active EAE where MOG35-55-specific T cells are activated in the draining

lymph nodes of the CFA /MOG deposit and from there migrate into the entire body of these mice including the lung, but also gut or meninges as shown by numerous publications. As induction of active EAE by the present approach also induces T cells specific for mycobacterial proteins and employs PTX and furthermore the present study unfortunately lacks a CFA control group the cellular and molecular underpinnings leading to CD45+ immune cell accumulation in the lung in this EAE model are very difficult to explain.

Thank you very much for this assessment and for stating that we provide convincing evidence for a role of Apelin/Apelin receptor interaction at the vascular level in the pathogenesis of EAE. Regarding the role of lung in this process, it is important to note that we do not rule out that the response to A13 treatment might involve other organs than the lung or vessel beds outside the pulmonary vasculature. Accordingly, we are also not making the claim that the lung is the only site where reprogramming of encephalitogenic T cells can occur. So far, however, we have no evidence for an involvement of other organs despite extensive analysis of different structures. The latest revision of our manuscript contains additional data intestine data and found that A13 treatment has no detectable effect on CD45+ cells (Suppl. Fig. 6a, b).

As the reviewer correctly points out, previous work by the team of Alexander FÃ¼gel had used transfer EAE in their landmark study that provided the first evidence for the reprogramming of encephalitogenic T cell programming in the lung (PMID: 22914092). We appreciate that transfer (passive) EAE avoids the systemic perturbation of the immune system and allows a more direct analysis of T cell function. But, like active EAE, the approach mimics only certain aspects of human multiple sclerosis (PMID: 36652203; PMID: 27766432). In the context of our manuscript, however, it is important to note that the team of

2

Alexander FÃ¼gel detected transferred (and labeled) T cells in recipient peripheral blood and spleen, which reflects that these cells have also access to the entire body and do not necessarily migrate towards the CNS without diversion. Duc et al. (PMID: 31597098), a reference provided by the reviewer in the next question (see below), show that adoptively transferred TCRMOG 2D2 Th17 cells proliferate in the colon. Thus, these studies have established that peripheral organs play important roles in neuroinflammation also in transfer EAE models. We have extended the Discussion and included additional references to reflect the relevant literature on this subject.

The study thus does not provide any direct evidence for the trafficking of encephalitogenic T cells via the lung in vivo and thus an effect of A13 treatment on ameliorating EAE due to delaying the trafficking of encephalitogenic T cells via the lung to the CNS.

The revised manuscript now includes additional timepoints investigating immune cell accumulation in the lung after EAE induction, however the authors still do not provide any data allowing to determine if these differences are due to immune cell entry, proliferation or trapping.

The revised study also does not sufficiently include investigation of encephalitogenic T cell trafficking and further activation in other mucosal sites such as the lamina propria of the small intestine and colon which are reported sites of encephalitogenic T cell activation (PMID: 37467267, doi.org/10.1016/j.celrep.2019.09.002) rather than the intestinal villi.

Thank you for your comments. With regard to the trafficking of encephalitogenic T cells via the lung, there is relevant existing literature and we are not the first to argue that lung plays a role in neuroinflammation. Apart from the work by Odoardi et al., which was already mentioned above (PMID: 22914092), Glenn et al. have proposed that the accumulation of granulocytic myeloid-derived suppressor cells in the lung might promote Th17 polarization and thereby the trafficking of autoimmune-targeted T cells through the lung (PMID: 30762897). Saito et al. showed that intratracheal delivery of nanoparticles can induce accumulation of CD4+ T cells in the lung rather than the CNS to prevent EAE pathology (PMID: 33067238). Likewise, Kanayama et al. have proposed that stalling of Th17 cells in the lung en route to the CNS delays EAE development. Nevertheless, while a search for the term "EAE" shows 12,886 results in Pubmed, the role of peripheral organs in neuroinflammation remains highly understudied.

Our own study benefits greatly from the fact that Apelin receptor (Aplnr) expression is restricted to a few organs in adult mice, which excludes the possibility that A13 acts directly on the CNS endothelium and thereby modulates, for example, the blood-brain-barrier. New data added to the revised manuscript (Suppl. Fig. 6a, b and below) shows the distribution of CD45+ cells throughout small intestine (including the lamina propria). These representative

3

images and our quantitation show no significant difference between control and A13-treated samples at different stages of EAE development.

Taken together, our manuscript provides substantial evidence indicating that A13 interferes with EAE development predominantly (but not necessarily exclusively) through the lung.

Finally, we would also like to emphasize that the accumulation of CD45+ cells in A13-treated EAE lungs is accompanied by a reduction of immune cells in peripheral blood and mediastinal lymph nodes, which drains the lung, at D11 (Suppl. Fig. 1b and 3b). Taken together, our data strongly argue that the effect of A13 on EAE progression involves the lung.

Due to the assumption of the central role of the lung in EAE pathogenesis the authors unfortunately limited themselves and their data interpretation solely on this organ and did not consider that A13 might also delay T cell entry into the CNS via the vasculature of the choroid plexus lacking a BBB or via the leptomeningeal vessels. In fact Supplementary Figure 5 shows a prominent GFP signal in the Aplnr-GFP-reporter mouse in the meninges which might reflect high Aplnr expression on meningeal vessels. The authors do not describe if there is a GFP reporter signal for Aplnr in the choroid plexus vasculature or in the meninges of the spinal cord.

Thank you very much for this comment even though we politely but firmly disagree that we were limited by preformed assumptions. After noting that A13 reduces EAE incidence, disease symptoms and immune cell number in the brain, we have analyzed a range of different organs and detected the most profound changes in the lung. In addition, our analysis was guided by the expression of Apelin receptor, which is very prominent in the lung and less so in other relevant organs. Aplnr-CreERT2-controlled GFP expression shows very limited labeling in CD31+ ECs (red) of the choroid plexus (see below).

4

This is fully consistent with a very low abundance of Aplnr+ cells in published single-nucleus RNA-seq data of the choroid plexus (PMID: 33932339), which we explored through the single cell portal of the Broad Institute (https://singlecell.broadinstitute.org/single_cell/study/SCP1366/choroid-plexus-nucleus-atlas). In the endothelial fraction (see below, left panel), expression is confined to two small clusters of embryonic nuclei, whereas the adult/aging endothelial (left) and epithelial clusters (right) are largely devoid of Aplnr+ nuclei. Given that A13 also did not lead to immune cell accumulation in this area, there is no good reason to believe that the effect of A13 is actually mediated by the choroid plexus.

In the leptomeninges, we see sparse GFP+ ECs (see Supplementary Fig. 5c and d) but no accumulation of immune cells in EAE, which is not altered by A13. In the dura mater, there is quite substantial Aplnr-CreERT2-induced GFP expression in CD31+ capillary ECs of young adult (8-week-old) mice (see below). Again, we do not see accumulation of CD45+ immune cells in response to EAE in this region. Moreover, there is no appreciable difference in CD45 immunostaining between vehicle control and A13-treated animals (see below).

5

These missing answers could be obtained by injecting a low number of fluorescently tagged naïve 2D2 cells prior to EAE induction and after CFA injection and following their migration in vivo into different organs. Alternatively, flow cytometry analysis of the immune cells accumulating in the lung employing MHC class II MOG35-55 tetramers would allow to determine the presence of encephalitogenic T cells in the lung versus other mucosal and meningeal sites. In vivo imaging of the lung and the brain/spinal cord would also allow to

obtain further insights.

We appreciate that the tracking of cells from naïve MOG-specific T cell receptor (TCR) transgenic might provide interesting insights into the migration of these cells into different organs in vivo. In fact, to our knowledge, this question has also not even been systematically addressed for transfer EAE. Likewise, in vivo imaging of lung and brain/spinal cord or other approaches might enhance our understanding of dynamic trafficking processes. At the same time, it needs to be considered that these approaches involve complex methods, are technically very demanding and have specific limitations. It is clear that such experiments extend far beyond the scope of the current manuscript and should be addressed in separate studies dedicated to T cell trafficking.

Technical concerns affecting data interpretation

Several immunostainings of the lung and the brain are very confusing. Figure 1d shows CD45 staining of lung sections – this staining should show a vast amount of alveolar macrophages already in a healthy lung, which are not visible. Similarly, Figure 1b depicts ICAM-1 staining on vascular structures but not on CD45+ immune cells, which also express high levels of ICAM-1 and thus should be brightly stained. Furthermore activated microglial cells stain positive for CD45 in brains of mice suffering from EAE – again this staining is not

6

visible in this figure. These examples raise concerns about the threshold settings during imaging or image analysis.

Thank you for this feedback. The two panels in question show low magnification overview images of lung sections. At the shown resolution, chosen for the visualization of comparably large CD45+ immune cell clusters, isolated individual cells are indeed not visible. Both images in Fig. 1d show lungs from EAE mice (vehicle and +A13) and therefore no healthy sample.

ICAM-1 immunostaining in EAE brain is most prominent in the endothelium. However, there is substantial ICAM-1 signal outside the vasculature, which includes CD45+ cells (see higher magnification of panel from Fig. 1b below).

Likewise, microglial cells are stained but not readily visible in the low magnification overview images provided in the manuscript.

The gating strategy shown in Supplementary Figure 21 lacks a figure legend and raises many questions on how the different immune cell subsets isolated from the different organs were analyzed and quantified throughout the manuscript. The gating strategy needs to be revised/corrected prior to reanalysis of the flow cytometry data. Specifically, a figure legend explaining the gating strategies is lacking and needs to be included. At present the gate referred to as lymphocyte gate includes 3 cellular subsets- what are these subsets? The CD45 gate to the right includes CD45 low cells, which is CNS resident microglial cells rather than CNS infiltrating immune cells – it seems the authors did not consider this population appropriately. The gate depicted as CD3 cells seems to be almost random as it depicts two populations with high CD3 cell surface expression, one population with very low CD3 cell surface expression, one negative population and one population with intermediate CD3 immunofluorescence that is cut in half by the shown gating strategy. It is also not clear if the gating shows a sequential gating. In panel C the lymphocyte gate is not a gate but rather a live cell gate, the CD4 gate depicts small and large cells with the larger CD4+ cells.

7

Thank you for these comments. We have improved the presentation of the gating strategy (see Suppl. Fig. 21 and 22) and reanalyzed our FACS results, which has led to minor adjustments in some graphs without changing outcomes or statistical significance.

Additional points:

Introduction:

The introduction still contains a number of inaccuracies with respect to MS, EAE and immune cell trafficking.

The etiology of MS is not known thus it is not appropriate to write "MS is mediated by autoreactive Th1 and Th17 cells but also involves B cells". In fact there is also evidence for a contribution of CD8 T cells in MS pathogenesis and environmental factors contributing to MS are not even mentioned. Furthermore it remains to be shown if autoaggressive T cells in MS see myelin antigens and EAE can also be induced with encephalitogenic T cells that do not recognize myelin antigens (see PMID: 30787438).

Thank you for this feedback. We have revised the introduction accordingly.

As outlined above the authors base their work on the assumption that it is also established that the generation of encephalitogenic T cells capable of penetrating the blood-brain barrier involves activation steps in lymph nodes, spleen and lung and research using EAE as an animal model has revealed an unexpected role of T cell reprogramming in the lung, which induces a migratory phenotype and enables the trafficking of these cells into the CNS. While in active EAE the priming, activation and first proliferation of encephalitogenic T cells in secondary lymphoid organs is a given the involvement of T cell activation steps in the lung is not an established concept in the sense that this is prerequisite for EAE pathogenesis. Rather, several studies have provided evidence for a gut-brain axis. The involvement of T cell reprogramming steps in the lung and mediastinal lymph nodes as prerequisite for their migration into the CNS has been observed in a transfer EAE model in Lewis rats where in vitro activated myelin specific T cell blasts are injected into syngeneic recipients. The model used in the present study is however very different as here encephalitogenic T cells are primed in vivo in the draining lymph nodes of the deposit of the CFA.

Thank you for these comments. We think that the role of the gastrointestinal system and the lack of evidence for an effect of A13 on this structure has been already discussed above. In essence, we are not at all contesting the existence of a gut-brain axis and are now citing a relevant reference (PMID: 31597098) in the introduction.

8

Furthermore, what is described as an "assumption" by the reviewer is actually a short recapitulation of previously published and directly relevant literature. Accordingly, the sentence on "T cell reprogramming in the lung" ends with two references provided in the bibliography of our manuscript.

Finally, the difference between active and transfer EAE was already discussed above.

The authors are advised to be more accurate when referring to T cells or other immune cell subsets in the context of immune cell trafficking. It is not correct that at the BBB $\alpha 4$ -integrin interactions with VCAM-1 replace the function of PSGL-1 and its selectin ligands. Selectins are upregulated at the inflamed BBB and mediate immune cell rolling (see in vivo studies of the Constantin, Kubes, Engelhardt laboratories). Integrins have been shown to mediate slow rolling but rather are involved in the subsequent steps of immune cell arrest, polarization and diapedesis. Also GPCR signaling induced by chemokine binding activates all integrins expressed by immune cells rather than specifically $\beta 2$ -integrins.

Thank you for this comment. We would like to point out politely that we did not actually state that " $\alpha 4$ -integrin interactions with VCAM-1 replace the function of PSGL-1 and its selectin ligands". However, to avoid any misunderstanding, we have modified the relevant sentences in the introduction.

Results- just several examples for further improvement given:

Several statements about EAE are incorrect in the manuscript. The authors eg. mention that C57BL/6 mice suffering from EAE are resistant to further induction of EAE as MOG35-55 induced EAE is a chronic disease one cannot induce EAE again in these mice as they will remain with mild symptoms until sacrificed. I guess the authors rather referred to the well known "adjuvans effect": it was indeed shown that EAE cannot be induced by active immunization in mice after CFA immunization. These observations highlight again why the CFA control in the present manuscript would be of utmost importance for data

interpretation.

A potential effect of A13 in relapsing-remitting or spontaneous EAE could be readily investigated in different EAE models available.

Again, we would like to point out that these statements, which were made to discuss limitations of the EAE model in the context of our work, reflect citations, in this case reference 68 and 69. Apart from these references, there is a very substantial body of literature on acquired resistance to EAE reintroduction (a few additional examples are PMID: 1690754, PMID: 2580957), but we appreciate that the same issue does not necessarily apply to transfer EAE (PMID: 6181092, PMID: 7327196). We also appreciate that there is a large variety of

9

different EAE models, all of which have their specific advantages and limitations. Future work will likely address the benefits of A13 in additional EAE models including those without active immunization.

The authors show that in vitro A13 inhibits T cell adhesion to the endothelium however they repeatedly mention that A13 interferes with leukocyte transendothelial migration â but there is no data showing that Aplnr regulates immune cell diapedesis across HUVECs or the vascular wall. Please note immune cell extravasation is a multi-step process in which each step is mediated by distinct molecules â thus the data should be reported accurately. Studying the effect of A13 on 2D2 T cell migration across mouse primary lung endothelial cells compared to brain microvascular endothelial cells would have been a very elegant way to show involvement of Aplnr in regulating encephalitogenic T cell trafficking across the lung versus brain endothelium.

Thank you very much for your comments. We obviously agree that leukocyte migration across the endothelium is a multistep process, which we have investigated with a live imaging approach in vitro (Fig. 5g, h) and transwell assays (Fig. 5i). We would also like to highlight the effect of A13 on VE-cadherin, which is a regulator of paracellular transendothelial migration.

To avoid any confusion, we have revised the sentence at the end of the relevant paragraph, which now reads: âTogether, these results show that Apelin reduces the transendothelial migration of T cells, but possibly also of other immune cells, which involves reduced adhesion to ECs.â

Figure 1 a: The authors refer to novel experiments with 8 mice per group in their reply to the reviewers â but this figure lacks error bars and mention of the number of animals investigated, the number of experiments performed and included into this figure. Thus it remains unclear if the study is underpowered and in addition reproducibility and variability of the observations described is not documented. These data need to be included and statistical analysis of a series of EAE experiments is best performed by comparing AUC for the clinical disease of all animals included using the statistical method of SuperPlots (Lord, S. J., Velle, K. B., Dyché Mullins, R. and Fritz-Laylin, L. K., Superplots: communicating reproducibility and variability in cell biology. *J. Cell Biol.* 2020. 219: e202001064.). All EAE experiments shown in this study should include provide this information.

Thank you for this suggestion. We have now combined all the animals from the two treatment groups and added error bars in Fig. 1a to make clear that our findings are based on

10

a robust number of animals and independent experiments (see Fig. 1a and below). The exact numbers are provided in the corresponding figure legend. The two-way ANOVA test between the control and A13-treated groups, which utilizes the average scores of individual experiments, shows statistical significance with a P value of less than 0.0001. The addition of error bars is not possible for the incidence data after D17 because this parameter corresponds to the number of affected animals so that error bars require data from multiple independent experiments. However, we only performed a single experiment with 8 mice in each group for the time period up to D30. This is because animals were sacrificed and analyzed at different

stages up to peak EAE and not kept until D30.

We appreciate that SuperPlots can provide a lot of relevant information (such as the distribution of individual data points, averages and color-coded replicates) within a single graph. Given that this can make graphs quite cluttered and difficult to read, we decided to keep the current format for data presentation.

Figure 1e is based on the Supplementary Figure 21 the gating strategies chosen for identifying immune cell subsets in the CNS have not considered CD45^{low} tissue resident microglial cells versus CD45^{high} infiltrating immune cells. These flow cytometry studies need to be reanalyzed and corrected.

Thank you. We have updated the FACS gating information (Suppl. Fig. 21 and 22) and, as mentioned already above, have reanalyzed the relevant data, which has led to minor adjustments in some graphs without changing outcomes or statistical significance.

Figure 2 states that A13 treatment downregulated ICAM-1 on endothelial cells but not epithelial cells – however the figure does not show any landmark allowing to determine ICAM-1 staining on epithelial cells.

11

Thank you very much for this comment. It is very well established that both pulmonary epithelial and endothelial cells express ICAM-1 (e.g. PMID: 8574890, PMID: 12108870) and epithelial cells lack expression of CD31. It is therefore straightforward to distinguish the two cell populations based on the staining provided in Fig. 2a. We have also marked the CD31⁺ endothelium with arrowheads. A13 reduces endothelial but not epithelial ICAM-1 expression in EAE mice.

Figure 5f – para and transcellular diapedesis cannot be distinguished without a landmark

Thank you for this comment. Cell-cell contact sites are visible in the bright field movies that have been used for the analysis (see below). The method is described in detail in PMID: 36382783.

Methods:

EAE scoring method used is not explained

Animals with EAE were scored daily for symptoms according to the EAE clinical severity scale: 0 = asymptomatic; 1 = partial loss of tail tonicity; 2 = tail paralysis; 3 = hind limb weakness; 4 = hind limb paralysis; 5 = 4-limb paralysis; 6 = death (PMID: 18541795).

Minor

The authors mix referring to leukocytes and immune cells – I propose to stick with the word “immune cell” as a summary as many researchers – and especially in the cell migration field use “leukocyte” to refer granulocytes.

Thank you for this comment. Presumably, we all agree that the term “leukocyte” (i.e., white blood cell) is actually correct and, looking through the published literature, the term “leukocyte” is frequently used in the context of EAE.

Orientation of tissue sections should be explained, e.g. orientation of the brain sections shown – are those coronal, transversal or sagittal sections.

These are sagittal sections, which is now indicated in the figure legends.

Version 2:

Reviewer comments:

Reviewer #4

(Remarks to the Author)

I thank the authors for their detailed reply to all the queries and adapting the manuscript accordingly. I would still caution on the role of Apelin in affecting the trafficking of encephalitogenic T cells through the lung as the study does not provide any direct evidence for this conclusion. Again solely the immunization with CFA will trigger an innate and adaptive immune response that has not been controlled for in this study lacking a CFA control group. Also the authors did not track MOG-specific T cells in any of their experiments. The study still does not show if the altered cell number of immune cells in the lung is due to higher numbers of cells trafficking there, local proliferation or longer dwelling times. I would thus suggest to downtone the manuscript in this regard.

Author Rebuttal letter:

REVIEWER COMMENTS

Reviewer #4 (Remarks to the Author):

I thank the authors for their detailed reply to all the queries and adapting the manuscript accordingly. I would still caution on the role of Apelin in affecting the trafficking of encephalitogenic T cells through the lung as the study does not provide any direct evidence for this conclusion. Again solely the immunization with CFA will trigger an innate and adaptive immune response that has not been controlled for in this study lacking a CFA control group. Also the authors did not track MOG-specific T cells in any of their experiments. The study still does not show if the altered cell number of immune cells in the lung is due to higher numbers of cells trafficking there, local proliferation or longer dwelling times. I would thus suggest to downtone the manuscript in this regard.

We thank the reviewer for the helpful feedback. The suggestions have been implemented in the final version of the article so that our conclusions are supported by the data.
